# Efficient Projection onto the Perfect Phylogeny Model

**Bei Jia**[*]   **Surjyendu Ray**                          **Sam Safavi**       **José Bento**
jiabe@bc.edu    raysc@bc.edu                         safavisa@bc.edu    jose.bento@bc.edu
                                    Boston College

## Abstract

Several algorithms build on the *perfect phylogeny model* to infer evolutionary trees. This problem is particularly hard when evolutionary trees are inferred from the fraction of genomes that have mutations in different positions, across different samples. Existing algorithms might do extensive searches over the space of possible trees. At the center of these algorithms is a projection problem that assigns a fitness cost to phylogenetic trees. In order to perform a wide search over the space of the trees, it is critical to solve this projection problem fast. In this paper, we use Moreau's decomposition for proximal operators, and a tree reduction scheme, to develop a new algorithm to compute this projection. Our algorithm terminates with an exact solution in a finite number of steps, and is extremely fast. In particular, it can search over all evolutionary trees with fewer than 11 nodes, a size relevant for several biological problems (more than 2 billion trees) in about 2 hours.

## 1   Introduction

The perfect phylogeny model (PPM) [1, 2] is used in biology to study evolving populations. It assumes that the same position in the genome never mutates twice, hence mutations only accumulate.

Consider a population of organisms evolving under the PPM. The evolution process can be described by a labeled rooted tree, $T = (r, \mathcal{V}, \mathcal{E})$, where $r$ is the root, i.e., the common oldest ancestor, the nodes $\mathcal{V}$ are the mutants, and the edges $\mathcal{E}$ are mutations acquired between older and younger mutants. Since each position in the genome only mutates once, we can associate with each node $v \neq r$, a unique mutated position, the mutation associated to the ancestral edge of $v$. By convention, let us associate with the root $r$, a null mutation that is shared by all mutants in $T$. This allows us to refer to each node $v \in \mathcal{V}$ as both a mutation in a position in the genome (the mutation associated to the ancestral edge of $v$), and a mutant (the mutant with the fewest mutations that has a mutation $v$). Hence, without loss of generality, $\mathcal{V} = \{1, \dots, q\}$, $\mathcal{E} = \{2, \dots, q\}$, where $q$ is the length of the genome, and $r = 1$ refers to both the oldest common ancestor and the null mutation shared by all.

One very important use of the PPM is to infer how mutants of a common ancestor evolve [3–8]. A common type of data used for this purpose is the frequency, with which different positions in the genome mutate across multiple samples, obtained, e.g., from whole-genome or targeted deep sequencing [9]. Consider a sample $s$, one of $p$ samples, obtained at a given stage of the evolution process. This sample has many mutants, some with the same genome, some with different genomes. Let $F \in \mathbb{R}^{q \times p}$ be such that $F_{v,s}$ is the fraction of genomes in $s$ with a mutation in position $v$ in the genome. Let $M \in \mathbb{R}^{q \times p}$ be such that $M_{v,s}$ is the fraction of mutant $v$ in $s$. By definition, the columns of $M$ must sum to 1. Let $U \in \{0,1\}^{q \times q}$ be such that $U_{v,v'} = 1$, if and only if mutant $v$ is an ancestor of mutant $v'$, or if $v = v'$. We denote the set of all possible $U$ matrices, $M$ matrices and labeled rooted trees $T$, by $\mathcal{U}$, $\mathcal{M}$ and $\mathcal{T}$, respectively. See Figure 1 for an illustration. The PPM implies

$$F = UM. \tag{1}$$

Our work contributes to the problem of inferring clonal evolution from mutation-frequencies: *How do we infer $M$ and $U$ from $F$?* Note that finding $U$ is the same as finding $T$ (see Lemma B.2).

---

[*]Bei Jia is currently with Element AI.

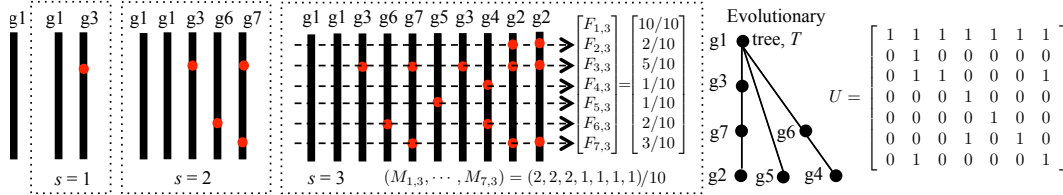

Figure 1: Black lines are genomes. Red circles indicate mutations. $gi$ is the mutant with fewest mutations with position $i$ mutated. Mutation 1, the mutation in the null position, $i = 1$, is shared by all mutants. $g1$ is the organism before mutant evolution starts. In sample $s = 3$, $2/10$ of the mutants are type $g2$, hence $M_{2,3} = 2/10$, and $3/10$ of the mutations occur in position 7, hence $F_{7,3} = 3/10$. The tree shows the mutants' evolution.

Although model (1) is simple, simultaneously inferring $M$ and $U$ from $F$ can be hard [3]. One popular inference approach is the following optimization problem over $U$, $M$ and $F$,

$$\min_{U \in \mathcal{U}} \mathcal{C}(U), \tag{2}$$

$$\mathcal{C}(U) = \min_{M,F \in \mathbb{R}^{q \times p}} \|\hat{F} - F\| \text{ subject to } F = UM, M \geq 0, M^\top \mathbf{1} = \mathbf{1}, \tag{3}$$

where $\|\cdot\|$ is the Frobenius norm, and $\hat{F} \in \mathbb{R}^{q \times p}$ contains the measured fractions of mutations per position in each sample, which are known and fixed. In a nutshell, we want to project our measurement $\hat{F}$ onto the space of valid PPM models.

Problem (2) is a hard mixed integer-continuous optimization problem. To approximately solve it, we might find a finite subset $\{U_i\} \subset \mathcal{U}$, that corresponds to a "heuristically good" subset of trees, $\{T_i\} \subset \mathcal{T}$, and, for each fixed matrix $U_i$, solve (3), which is a convex optimization problem. We can then return $T_x$, where $x \in \arg\min_i \mathcal{C}(U_i)$. Fortunately, in many biological applications, e.g., [3–8], the reconstructed evolutionary tree involves a very small number of mutated positions, e.g., $q \leq 11$. In practice, a position $v$ might be an *effective* position that is a cluster of multiple real positions in the genome. For a small $q$, we can compute $\mathcal{C}(U)$ for many trees, and hence approximate $M, U$, and get uncertainty measures for these estimates. This is important, since data is generally scarce and noisy.

**Contributions:** (i) we propose a new algorithm to compute $\mathcal{C}(U)$ exactly in $\mathcal{O}(q^2 p)$ steps, the first non-iterative algorithm to compute $\mathcal{C}(U)$; (ii) we compare its performance against state-of-the-art iterative algorithms, and observe a much faster convergence. In particular, our algorithm scales much faster than $\mathcal{O}(q^2 p)$ in practice; (iii) we implement our algorithm on a GPU, and show that it computes the cost of all (more than 2 billion) trees with $\leq 11$ nodes, in $\leq 2.5$ hours.

## 2 Related work

A problem related to ours, but somewhat different, is that of inferring a phylogenetic tree from single-cell whole-genome sequencing data. Given all the mutations in a set of mutants, the problem is to arrange the mutants in a phylogenetic tree, [10, 11]. Mathematically, this corresponds to inferring $T$ from partial or corrupted observation of $U$. If the PPM is assumed, and all the mutations of all the mutants are correctly observed, this problem can be solved in linear time, e.g., [12]. In general, this problem is equivalent to finding a minimum cost Steiner tree on a hypercube, whose nodes and edges represent mutants and mutations respectively, a problem known to be hard [13].

We mention a few works on clonality inference, based on the PPM, that try to infer both $U$ and $M$ from $\hat{F}$. No previous work solves problem (2) exactly in general, even for trees of size $q \leq 11$. Using our fast projection algorithm, we can solve (2) exactly by searching over all trees, if $q \leq 11$. Ref. [3] (*AncesTree*) reduces the space of possible trees $\mathcal{T}$ to subtrees of a heuristically constructed DAG. The authors use the element-wise 1-norm in (3) and, after introducing more variables to linearize the product $UM$, reduce this search to solving a MILP, which they try to solve via branch and bound. Ref. [6] (*CITUP*) searches the space of all *unlabeled* trees, and, for each unlabeled tree, tries to solve an MIQP, again using branch and bound techniques, which finds a labeling for the unlabeled tree, and simultaneously minimizes the distance $\|\hat{F} - F\|$. Refs. [5] and [14] (*PhyloSub/PhyloWGS*), use a stochastic model to sample trees that are likely to explain the data. Their model is based on [15], which generates hierarchical clusterings of objects, and from which lineage trees can be formed. A score is then computed for these trees, and the highest scoring trees are returned.

Procedure (2) can be justified as MLE if we assume the stochastic model $\hat{F} = F + \mathcal{N}(0, I\sigma^2)$, where $F$, $U$ and $M$ satisfy the PPM model, and $\mathcal{N}(0, I\sigma^2)$ represents additive, component-wise,

Gaussian measurement noise, with zero mean and covariance $I\sigma^2$. Alternative stochastic models can be assumed, e.g., as $M - U^{-1}\hat{F} = \mathcal{N}(0, I\sigma^2)$, where $M$ is non-negative and its columns must sum to one, and $\mathcal{N}(0, I\sigma^2)$ is as described before. For this model, and for each matrix $U$, the cost $\mathcal{C}(U)$ is a projection of $U^{-1}\hat{F}$ onto the probability simplex $M \geq 0, M^\top \mathbf{1} = \mathbf{1}$. Several fast algorithms are known for this problem, e.g., [16–20] and references therein. In a $pq$-dimensional space, the exact projection onto the simplex can be done in $\mathcal{O}(qp)$ steps.

Our algorithm is the first to solve (3) exactly in a finite number of steps. We can also use iterative methods to solve (3). One advantage of our algorithm is that it has no tuning parameters, and requires no effort to check for convergence for a given accuracy. Since iterative algorithms can converge very fast, we numerically compare the speed of our algorithm with different implementations of the Alternating Direction Method of Multipliers (ADMM) [21], which, if properly tuned, has a convergence rate that equals the fastest convergence rate among all first order methods [22] under some convexity assumptions, and is known to produce good solutions for several other kinds of problems, even for non-convex ones [23–29].

## 3 Main results

We now state our main results, and explain the ideas behind their proofs. Detailed proofs can be found in the Appendix.

Our algorithm computes $\mathcal{C}(U)$ and minimizers of (3), resp. $M^*$ and $F^*$, by solving an equivalent problem. Without loss of generality, we assume that $p = 1$, since, by squaring the objective in (3), it decomposes into $p$ independent problems. Sometimes we denote $\mathcal{C}(U)$ by $\mathcal{C}(T)$, since given $U$, we can specify $T$, and vice-versa. Let $\bar{i}$ be the closest ancestor of $i$ in $T = (r, \mathcal{V}, \mathcal{E})$. Let $\Delta i$ be the set of all the ancestors of $i$ in $T$, plus $i$. Let $\partial i$ be the set of children of $i$ in $T$.

**Theorem 3.1** (Equivalent formulation). *Problem* (3) *can be solved by solving*

$$\min_{t \in \mathbb{R}} \quad t + \mathcal{L}(t), \tag{4}$$
$$\mathcal{L}(t) = \min_{Z \in \mathbb{R}^q} \frac{1}{2} \sum_{i \in \mathcal{V}} (Z_i - Z_{\bar{i}})^2 \text{ subject to } Z_i \leq t - N_i \,, \forall i \in \mathcal{V}, \tag{5}$$

*where $N_i = \sum_{j \in \Delta i} \hat{F}_j$, and, by convention, $Z_{\bar{i}} = 0$ for $i = r$. In particular, if $t^*$ minimizes* (4), *$Z^*$ minimizes* (5) *for $t = t^*$, and $M^*$, $F^*$ minimize* (3), *then*

$$M_i^* = -Z_i^* + Z_{\bar{i}}^* + \sum_{r \in \partial i} (Z_r^* - Z_{\bar{r}}^*) \text{ and } F_i^* = -Z_i^* + Z_{\bar{i}}^*, \forall i \in \mathcal{V}. \tag{6}$$

*Furthermore, $t^*$, $M^*$, $F^*$ and $Z^*$ are unique.*

Theorem 3.1 comes from a dual form of (3), which we build using Moreau's decomposition [30].

### 3.1 Useful observations

Let $Z^*(t)$ be the unique minimizer of (5) for some $t$. The main ideas behind our algorithm depend on a few simple properties of the paths $\{Z^*(t)\}$ and $\{\mathcal{L}'(t)\}$, the derivative of $\mathcal{L}(t)$ with respect to $t$. Note that $\mathcal{L}$ is also a function of $N$, as defined in Theorem 3.1, which depends on the input data $\hat{F}$.

**Lemma 3.2.** *$\mathcal{L}(t)$ is a convex function of $t$ and $N$. Furthermore, $\mathcal{L}(t)$ is continuous in $t$ and $N$, and $\mathcal{L}'(t)$ is non-decreasing with $t$.*

**Lemma 3.3.** *$Z^*(t)$ is continuous as a function of $t$ and $N$. $Z^*(t^*)$ is continuous as a function of $N$.*

Let $\mathcal{B}(t) = \{i : Z^*(t)_i = t - N_i\}$, i.e., the set of components of the solution at the boundary of (5). Variables in $\mathcal{B}$ are called *fixed*, and we call other variables *free*. Free (resp. fixed) nodes are nodes corresponding to free (resp. fixed) variables.

**Lemma 3.4.** *$\mathcal{B}(t)$ is piecewise constant in $t$.*

Consider dividing the tree $T = (r, \mathcal{V}, \mathcal{E})$ into subtrees, each with at least one free node, using $\mathcal{B}(t)$ as separation points. See Figure 4 in Appendix A for an illustration. Each $i \in \mathcal{B}(t)$ belongs to at most degree$(i)$ different subtrees, where degree$(i)$ is the degree of node $i$, and each $i \in \mathcal{V} \backslash \mathcal{B}(t)$ belongs exactly to one subtree. Let $T_1, \ldots, T_k$ be the set of resulting (rooted, labeled) trees. Let $T_w = (r_w, \mathcal{V}_w, \mathcal{E}_w)$, where the root $r_w$ is the closest node in $T_w$ to $r$. We call $\{T_w\}$ the subtrees *induced by* $\mathcal{B}(t)$. We define $\mathcal{B}_w(t) = \mathcal{B}(t) \cap \mathcal{V}_w$, and, when it does not create ambiguity, we drop the index $t$ in $\mathcal{B}_w(t)$. Note that different $\mathcal{B}_w(t)$'s might have elements in common. Also note that, by construction, if $i \in \mathcal{B}_w$, then $i$ must be a leaf of $T_w$, or the root of $T_w$.

**Definition 3.5.** The $(T_w, \mathcal{B}_w)$-problem is the optimization problem over $|\mathcal{V}_w \backslash \mathcal{B}(t)|$ variables

$$\min_{\{Z_j : j \in \mathcal{V}_w \backslash \mathcal{B}(t)\}} (1/2) \sum_{j \in \mathcal{V}_w} (Z_j - Z_{\bar{j}})^2, \tag{7}$$

where $\bar{j}$ is the parent of $j$ in $T_w$, $Z_{\bar{j}} = 0$ if $j = r_w$, and $Z_j = Z^*(t)_j = t - N_j$ if $j \in \mathcal{B}_w(t)$.

**Lemma 3.6.** *Problem* (5) *decomposes into $k$ independent problems. In particular, the minimizers $\{Z^*(t)_j : j \in \mathcal{V}_w \backslash \mathcal{B}(t)\}$ are determined as the solution of the $(T_w, \mathcal{B}_w)$-problem. If $j \in \mathcal{V}_w$, then $Z^*(t)_j = c_1 t + c_2$, where $c_1$ and $c_2$ depend on $j$ but not on $t$, and $0 \leq c_1 \leq 1$.*

**Lemma 3.7.** *$Z^*(t)$ and $\mathcal{L}'(t)$ are piecewise linear and continuous in $t$. Furthermore, $Z^*(t)$ and $\mathcal{L}'(t)$ change linear segments if and only if $\mathcal{B}(t)$ changes.*

**Lemma 3.8.** *If $t \leq t'$, then $\mathcal{B}(t') \subseteq \mathcal{B}(t)$. In particular, $\mathcal{B}(t)$ changes at most $q$ times with $t$.*

**Lemma 3.9.** *$Z^*(t)$ and $\mathcal{L}'(t)$ have less than $q + 1$ different linear segments.*

## 3.2 The Algorithm

In a nutshell, our algorithm computes the solution path $\{Z^*(t)\}_{t \in \mathbb{R}}$ and the derivative $\{\mathcal{L}'(t)\}_{t \in \mathbb{R}}$. From these paths, it finds the unique $t^*$, at which

$$\mathrm{d}(t + \mathcal{L}(t))/\mathrm{d}t = 0|_{t=t^*} \Leftrightarrow \mathcal{L}'(t^*) = -1. \tag{8}$$

It then evaluates the path $Z^*(t)$ at $t = t^*$, and uses this value, along with (6), to find $M^*$ and $F^*$, the unique minimizers of (3). Finally, we compute $\mathcal{C}(T) = \|\hat{F} - F^*\|$.

We know that $\{Z^*(t)\}$ and $\{\mathcal{L}'(t)\}$ are continuous piecewise linear, with a finite number of different linear segments (Lemmas 3.7, 3.8 and 3.9). Hence, to describe $\{Z^*(t)\}$ and $\{\mathcal{L}'(t)\}$, we only need to evaluate them at the *critical values*, $t_1 > t_2 > \cdots > t_k$, at which $Z^*(t)$ and $\mathcal{L}'(t)$ change linear segments. We will later use Lemma 3.7 as a criteria to find the critical values. Namely, $\{t_i\}$ are the values of $t$ at which, as $t$ decreases, new variables become fixed, and $\mathcal{B}(t)$ changes. Note that variables never become free once fixed, by Lemma 3.8, which also implies that $k \leq q$.

The values $\{Z^*(t_i)\}$ and $\{\mathcal{L}'(t_i)\}$ are computed sequentially as follows. If $t$ is very large, the constraint in (5) is not active, and $Z^*(t) = \mathcal{L}(t) = \mathcal{L}'(t) = 0$. Lemma 3.7 tells us that, as we decrease $t$, the first critical value is the largest $t$ for which this constraint becomes active, and at which $\mathcal{B}(t)$ changes for the first time. Hence, if $i = 1$, we have $t_i = \max_s\{N_s\}$, $Z^*(t_i) = \mathcal{L}'(t_i) = 0$, and $\mathcal{B}(t_i) = \arg\max_s\{N_s\}$. Once we have $t_i$, we compute the rates $Z'^*(t_i)$ and $\mathcal{L}''(t_i)$ from $\mathcal{B}(t_i)$ and $T$, as explained in Section 3.3. Since the paths are piecewise linear, derivatives are not defined at critical points. Hence, here, and throughout this section, these derivatives are taken from the left, i.e., $Z'^*(t_i) = \lim_{t \uparrow t_i}(Z^*(t_i) - Z^*(t))/(t_i - t)$ and $\mathcal{L}''(t_i) = \lim_{t \uparrow t_i}(\mathcal{L}'(t_i) - \mathcal{L}'(t))/(t_i - t)$.

Since $Z'^*(t)$ and $\mathcal{L}''(t)$ are constant for $t \in (t_{i+1}, t_i]$, for $t \in (t_{i+1}, t_i]$ we have

$$Z^*(t) = Z^*(t_i) + (t - t_i)Z'^*(t_i), \quad \mathcal{L}'(t) = \mathcal{L}'(t_i) + (t - t_i)\mathcal{L}''(t_i), \tag{9}$$

and the next critical value, $t_{i+1}$, is the largest $t < t_i$, for which new variables become fixed, and $\mathcal{B}(t)$ changes. The value $t_{i+1}$ is found by solving for $t < t_i$ in

$$Z^*(t)_r = Z^*(t_i)_r + (t - t_i)Z'^*(t_i)_r = t - N_r, \tag{10}$$

and keeping the largest solution among all $r \notin \mathcal{B}$. Once $t_{i+1}$ is computed, we update $\mathcal{B}$ with the new variables that became fixed, and we obtain $Z^*(t_{i+1})$ and $\mathcal{L}'(t_{i+1})$ from (9). The process then repeats.

By Lemma 3.2, $\mathcal{L}'$ never increases. Hence, we stop this process (a) as soon as $\mathcal{L}'(t_i) < -1$, or (b) when all the variables are in $\mathcal{B}$, and thus there are no more critical values to compute. If (a), let $t_k$ be the last critical value with $\mathcal{L}'(t_k) > -1$, and if (b), let $t_k$ be the last computed critical value. We use $t_k$ and (9) to compute $t^*$, at which $\mathcal{L}'(t^*) = -1$ and also $Z^*(t^*)$. From $Z^*(t^*)$ we then compute $M^*$ and $F^*$ and $\mathcal{C}(U) = \|\hat{F} - F^*\|$.

The algorithm is shown compactly in Alg. 1. Its inputs are $\hat{F}$ and $T$, represented, e.g., using a linked-nodes data structure. Its outputs are minimizers to (3). It makes use of a procedure *ComputeRates*, which we will explain later. This procedure terminates in $\mathcal{O}(q)$ steps and uses $\mathcal{O}(q)$ memory. Line 5 comes from solving (10) for $t$. In line 14, the symbols $M^*(Z^*, T)$ and $F^*(Z^*, T)$ remind us that $M^*$ and $F^*$ are computed from $Z^*$ and $T$ using (6). The correctness of Alg. 1 follows from the Lemmas in Section 3.1, and the explanation above. In particular, since there are at most $q + 1$ different linear regimes, the bound $q$ in the for-loop does not prevent us from finding any critical value. Its time complexity is $\mathcal{O}(q^2)$, since each line completes in $\mathcal{O}(q)$ steps, and is executed at most $q$ times.

**Theorem 3.10** (Complexity). *Algorithm 1 finishes in $\mathcal{O}(q^2)$ steps, and requires $\mathcal{O}(q)$ memory.*

**Theorem 3.11** (Correctness). *Algorithm 1 outputs the solution to* (3).

---

**Algorithm 1** Projection onto the PPM (input: $T$ and $\hat{F}$; output: $M^*$ and $F^*$)

---

1: $N_i = \sum_{j \in \Delta_i} \hat{F}_j$, for all $i \in \mathcal{V}$ ▷ This takes $\mathcal{O}(q)$ steps using a DFS, see proof of Theorem 3.10
2: $i = 1, t_i = \max_r\{N_r\}, \mathcal{B}(t_i) = \arg\max_r\{N_r\}, Z^*(t_i) = \mathbf{0}, \mathcal{L}'(t_i) = 0.$      ▷ Initialize
3: **for** $i = 1$ to $q$ **do**
4:      $(Z'^*(t_i), \mathcal{L}''(t_i)) = \text{ComputeRates}(\mathcal{B}(t_i), T)$      ▷ Update rates of change
5:      $P = \{P_r : P_r = \frac{N_r + Z^*(t_i)_r - t_i Z'^*(t_i)_r}{1 - Z'^*(t_i)_r}$ if $r \notin \mathcal{B}(t_i), t_r < t_i$, and $P_r = -\infty$ otherwise$\}$
6:      $t_{i+1} = \max_r P_r$      ▷ Update next critical value from (9)
7:      $\mathcal{B}(t_{i+1}) = \mathcal{B}(t_i) \cup \arg\max_r P_s$      ▷ Update list of fixed variables
8:      $Z^*(t_{i+1}) = Z^*(t_i) + (t_{i+1} - t_i)Z'^*(t_i)$      ▷ Update solution path
9:      $\mathcal{L}'(t_{i+1}) = \mathcal{L}'(t_i) + (t_{i+1} - t_i)\mathcal{L}''(t_i)$      ▷ Update objective's derivative
10:      **if** $\mathcal{L}'(t_{i+1}) < -1$ **then break**      ▷ If already passed by $t^*$, then exit the for-loop
11: **end for**
12: $t^* = t_i - \frac{1 + \mathcal{L}'(t_i)}{\mathcal{L}''(t_i)}$      ▷ Find solution to (8)
13: $Z^* = Z^*(t_i) + (t^* - t_i)Z'^*(t_i)$      ▷ Find minimizers of (5) for $t = t^*$
14: **return** $M^*(Z^*, T), F^*(Z^*, T)$      ▷ Return solution to (3) using (6), which takes $\mathcal{O}(q)$ steps

---

### 3.3 Computing the rates

We now explain how the procedure *ComputeRates* works. Recall that it takes as input the tree $T$ and the set $\mathcal{B}(t_i)$, and it outputs the derivatives $Z'^*(t_i)$ and $\mathcal{L}''(t_i)$.

A simple calculation shows that if we compute $Z'^*(t_i)$, then computing $\mathcal{L}''(t_i)$ is easy.

**Lemma 3.12.** *$\mathcal{L}''(t_i)$ can be computed from $Z'^*(t_i)$ in $\mathcal{O}(q)$ steps and with $\mathcal{O}(1)$ memory as*

$$\mathcal{L}''(t_i) = \sum_{j \in \mathcal{V}} (Z'^*(t_i)_j - Z'^*(t_i)_{\bar{j}})^2, \tag{11}$$

where $\bar{j}$ is the closest ancestor to $j$ in $T$. We note that if $j \in \mathcal{B}(t_i)$, then, by definition, $Z'^*(t_i)_j = 1$. Assume now that $j \in \mathcal{V} \backslash \mathcal{B}(t_i)$. Lemma 3.6 implies we can find $Z'^*(t_i)_j$ by solving the $(T_w = (r_w, \mathcal{V}_w, \mathcal{E}_w), \mathcal{B}_w)$-problem as a function of $t$, where $w$ is such that $j \in \mathcal{V}_w$. In a nutshell, *ComputeRates* is a recursive procedure to solve all the $(T_w, \mathcal{B}_w)$-problems as an explicit function of $t$.

It suffices to explain how *ComputeRates* solves one particular $(T_w, \mathcal{B}_w)$-problem explicitly. To simplify notation, in the rest of this section, we refer to $T_w$ and $\mathcal{B}_w$ as $T$ and $\mathcal{B}$. Recall that, by the definition of $T = T_w$ and $\mathcal{B} = \mathcal{B}_w$, if $i \in \mathcal{B}$, then $i$ must be a leaf of $T$, or the root of $T$.

**Definition 3.13.** Consider a rooted tree $T = (r, \mathcal{V}, \mathcal{E})$, a set $\mathcal{B} \subseteq \mathcal{V}$, and variables $\{Z_j : j \in \mathcal{V}\}$ such that, if $j \in \mathcal{B}$, then $Z_j = \alpha_j t + \beta_j$ for some $\alpha$ and $\beta$. We define the $(T, \mathcal{B}, \alpha, \beta, \gamma)$-problem as

$$\min_{\{Z_j : j \in \mathcal{V} \backslash \mathcal{B}\}} \frac{1}{2} \sum_{j \in \mathcal{V}} \gamma_j (Z_j - Z_{\bar{j}})^2, \tag{12}$$

where $\gamma > 0$, $\bar{j}$ is the closest ancestor to $j$ in $T$, and $Z_{\bar{j}} = 0$ if $j = r$.

We refer to the solution of the $(T, \mathcal{B}, \alpha, \beta, \gamma)$-problem as $\{Z_j^* : j \in \mathcal{V} \backslash \mathcal{B}\}$, which uniquely minimizes (12). Note that (12) is unconstrained and its solution, $Z^*$, is a linear function of $t$. Furthermore, the $(T_w, \mathcal{B}_w)$-problem is the same as the $(T_w, \mathcal{B}_w, \mathbf{1}, -N, \mathbf{1})$-problem, which is what we actually solve.

We now state three useful lemmas that help us solve any $(T, \mathcal{B}, \alpha, \beta, \gamma)$-problem efficiently.

**Lemma 3.14** (Pruning). *Consider the solution $Z^*$ of the $(T, \mathcal{B}, \alpha, \beta, \gamma)$-problem. Let $j \in \mathcal{V} \backslash \mathcal{B}$ be a leaf. Then $Z_j^* = Z_{\bar{j}}^*$. Furthermore, consider the $(\tilde{T}, \mathcal{B}, \alpha, \beta, \gamma)$-problem, where $\tilde{T} = (\tilde{r}, \tilde{\mathcal{V}}, \tilde{\mathcal{E}})$ is equal to $T$ with node $j$ pruned, and let its solution be $\tilde{Z}^*$. We have that $Z_i^* = \tilde{Z}_i^*$, for all $i \in \tilde{\mathcal{V}}$.*

**Lemma 3.15** (Star problem). *Let $T$ be a star such that node 1 is the center node, node 2 is the root, and nodes $3, \ldots, r$ are leaves. Let $\mathcal{B} = \{2, \ldots, r\}$. Let $Z_1^* \in \mathbb{R}$ be the solution of the $(T, \mathcal{B}, \alpha, \beta, \gamma)$-problem. Then,*

$$Z_1^* = \left( \frac{\gamma_1 \alpha_2 + \sum_{i=3}^{r} \gamma_r \alpha_r}{\gamma_1 + \sum_{i=3}^{r} \gamma_r} \right) t + \left( \frac{\gamma_1 \beta_2 + \sum_{i=3}^{r} \gamma_r \beta_r}{\gamma_1 + \sum_{i=3}^{r} \gamma_r} \right). \tag{13}$$

*In particular, to find the rate at which $Z_1^*$ changes with $t$, we only need to know $\alpha$ and $\gamma$, not $\beta$.*

**Lemma 3.16** (Reduction). *Consider the $(T, \mathcal{B}, \alpha, \beta, \gamma)$-problem such that $j, \bar{j} \in \mathcal{V} \backslash \mathcal{B}$, and such that $j$ has all its children $1, \ldots, r \in \mathcal{B}$. Let $Z^*$ be its solution. Consider the $(\tilde{T}, \tilde{\mathcal{B}}, \tilde{\alpha}, \tilde{\beta}, \tilde{\gamma}) - problem$, where $\tilde{T} = (\tilde{r}, \tilde{\mathcal{V}}, \tilde{\mathcal{E}})$ is equal to $T$ with nodes $1, \ldots, r$ removed, and $\tilde{\mathcal{B}} = (\mathcal{B} \backslash \{1, \ldots, r\}) \cup \{j\}$. Let $\tilde{Z}^*$ be its solution. If $(\tilde{\alpha}_i, \tilde{\beta}_i, \tilde{\gamma}_i) = (\alpha_i, \beta_i, \gamma_i)$ for all $i \in \mathcal{B} \backslash \{1, \ldots, r\}$, and $\tilde{\alpha}_j$, $\tilde{\beta}_j$ and $\tilde{\gamma}_j$ satisfy*

$$\tilde{\alpha}_j = \frac{\sum_{i=1}^r \gamma_i \alpha_i}{\sum_{i=1}^r \gamma_i}, \quad \tilde{\beta}_j = \frac{\sum_{i=1}^r \gamma_i \beta_i}{\sum_{i=1}^r \gamma_i}, \tilde{\gamma}_j = \left( (\gamma_j)^{-1} + \left( \sum_{i=1}^r \gamma_i \right)^{-1} \right)^{-1}, \quad (14)$$

*then $Z^*_i = \tilde{Z}^*{}_i$ for all $i \in \mathcal{V} \backslash \{j\}$.*

Lemma 3.15 and Lemma 3.16 allow us to recursively solve any $(T, \mathcal{B}, \alpha, \beta, \gamma)$-problem, and obtain for it an explicit solution of the form $Z^*(t) = c_1 t + c_2$, where $c_1$ and $c_2$ do not depend on $t$.

Assume that we have already repeatedly pruned $T$, by repeatedly invoking Lemma 3.14, such that, if $i$ is a leaf, then $i \in \mathcal{B}$. See Figure 2-(left). First, we find some node $j \in \mathcal{V} \backslash \mathcal{B}$ such that all of its children are in $\mathcal{B}$. If $\bar{j} \in \mathcal{B}$, then $\bar{j}$ must be the root, and the $(T, \mathcal{B}, \alpha, \beta, \gamma)$-problem must be a star problem as in Lemma 3.15. We can use Lemma 3.15 to solve it explicitly. Alternatively, if $\bar{j} \notin \mathcal{V} \backslash \mathcal{B}$, then we invoke Lemma 3.16, and reduce the $(T, \mathcal{B}, \alpha, \beta, \gamma)$-problem to a strictly smaller $(\tilde{T}, \tilde{\mathcal{B}}, \tilde{\alpha}, \tilde{\beta}, \tilde{\gamma})$-problem, which we solve recursively. Once the $(\tilde{T}, \tilde{\mathcal{B}}, \tilde{\alpha}, \tilde{\beta}, \tilde{\gamma})$-problem is solved, we have an explicit expression $Z^*_i(t) = c_{1i} t + c_{2i}$ for all $i \in \mathcal{V} \backslash \{j\}$, and, in particular, we have an explicit expression $Z^*_{\bar{j}}(t) = c_{1\bar{j}} t + c_{2\bar{j}}$. The only free variable of the $(T, \mathcal{B}, \alpha, \beta, \gamma)$-problem to be determined is $Z^*_j(t)$. To compute $Z^*_j(t)$, we apply Lemma 3.15 to the $(\tilde{\tilde{T}}, \tilde{\tilde{\mathcal{B}}}, \tilde{\tilde{\alpha}}, \tilde{\tilde{\beta}}, \tilde{\tilde{\gamma}})$-problem, where $\tilde{\tilde{T}}$ is a star around $j$, $\tilde{\tilde{\gamma}}$ are the components of $\gamma$ corresponding to nodes that are neighbors of $j$, $\tilde{\tilde{\alpha}}$ and $\tilde{\tilde{\beta}}$ are such that $Z^*_i(t) = \tilde{\tilde{\alpha}}_i t + \tilde{\tilde{\beta}}_i$ for all $i$ that are neighbors of $j$, and for which $Z^*_i(t)$ is already known, and $\tilde{\tilde{\mathcal{B}}}$ are all the neighbors of $j$. See Figure 2-(right).

The algorithm is compactly described in Alg. 2. It is slightly different from the description above for computational efficiency. Instead of computing $Z^*(t) = c_1 t + c_2$, we keep track only of $c_1$, the rates, and we do so only for the variables in $\mathcal{V} \backslash \mathcal{B}$. The algorithm assumes that the input $T$ has been pruned. The inputs $T$, $\mathcal{B}$, $\alpha$, $\beta$ and $\gamma$ are passed by reference. They are modified inside the algorithm but, once *ComputeRatesRec* finishes, they keep their initial values. Throughout the execution of the algorithm, $T = (r, \mathcal{V}, \mathcal{E})$ encodes (1) a doubly-linked list where each node points to its children and its parent, which we call $T.a$, and (b) a a doubly-linked list of all the nodes in $\mathcal{V} \backslash \mathcal{B}$ for which all the children are in $\mathcal{B}$, which we call $T.b$. In the proof of Theorem 3.17, we prove how this representation of $T$ can be kept updated with little computational effort. The input $Y$, also passed by reference, starts as an uninitialized array of size $q$, where we will store the rates $\{Z'^*_i\}$. At the end, we read $Z'^*$ from $Y$.

---

**Algorithm 2** ComputeRatesRec (input: $T = (r, \mathcal{V}, \mathcal{E}), \mathcal{B}, \alpha, \beta, \gamma, Y$)

---

1: Let $j$ be some node in $\mathcal{V} \backslash \mathcal{B}$ whose children are in $\mathcal{B}$        ▷ We read $j$ from $T.b$ in $\mathcal{O}(1)$ steps
2: **if** $\bar{j} \in \mathcal{B}$ **then**
3:     Set $Y_j$ using (13) in Lemma 3.15 ▷ If $\bar{j} \in \mathcal{B}$, then the $(T, \mathcal{B}, \alpha, \beta, \gamma)$-problem is star-shaped
4: **else**
5:     Modify $(T, \mathcal{B}, \alpha, \beta, \gamma)$ to match $(\tilde{T}, \tilde{\mathcal{B}}, \tilde{\alpha}, \tilde{\beta}, \tilde{\gamma})$ defined by Lemma 3.16 for $j$ in line 1
6:     ComputeRatesRec$(T, \mathcal{B}, \alpha, \beta, \gamma, Y)$   ▷ Sets $Y_i = Z'^*_i$ for all $i \in \mathcal{V} \backslash \mathcal{B}$; $Y_j$ is not yet defined
7:     Restore $(T, \mathcal{B}, \alpha, \beta, \gamma)$ to its original value before line 5 was executed
8:     Compute $Y_j$ from (13), using for $\alpha, \beta, \gamma$ in (13) the values $\tilde{\tilde{\alpha}}, \tilde{\tilde{\beta}}, \tilde{\tilde{\gamma}}$, where $\tilde{\tilde{\gamma}}$ are the components of $\gamma$ corresponding to nodes that are neighbors of $j$ in $T$, and $\tilde{\tilde{\alpha}}$ and $\tilde{\tilde{\beta}}$ are such that $Z^*_i = \tilde{\tilde{\alpha}}_i t + \tilde{\tilde{\beta}}_i$ for all $i$ that are neighbors of $j$ in $T$, and for which $Z^*_i$ is already known
9: **end if**

---

Let $q$ be the number of nodes of the tree $T$ that is the input at the zeroth level of the recursion.

**Theorem 3.17.** *Algorithm 2 correctly computes $Z'^*$ for the $(T, \mathcal{B}, \alpha, \beta, \gamma)$-problem, and it can be implemented to finish in $\mathcal{O}(q)$ steps, and to use $\mathcal{O}(q)$ memory.*

The correctness of Algorithm 2 follows from Lemmas 3.14-3.16, and the explanation above. Its complexity is bounded by the total time spent on the two lines that actually compute rates during

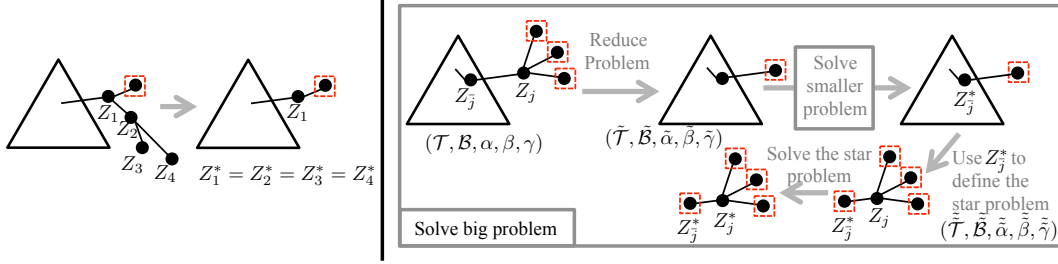

Figure 2: Red squares represent fixed nodes, and black circles free nodes. (Left) By repeatedly invoking Lemma 3.14, we can remove nodes 2, 3, and 4 from the original problem, since their associated optimal values are equal to the optimal value for node 1. (Right) We can compute the rates for all the free nodes of a subtree recursively by applying Lemma 3.16 and Lemma 3.15. We know the linear behavior of variables associated to red squares.

the whole recursion, lines 3 and 8. All the other lines only transform the input problem into a more computable form. Lines 3 and 8 solve a star-shaped problem with at most $degree(j)$ variables, which, by inspecting (13), we know can be done in $\mathcal{O}(\mathrm{degree}(j))$ steps. Since, $j$ never takes the same value twice, the overall complexity is bounded by $\mathcal{O}(\sum_{j \in \mathcal{V}} \mathrm{degree}(j)) = \mathcal{O}(|\mathcal{E}|) = \mathcal{O}(q)$. The $\mathcal{O}(q)$ bound on memory is possible because all the variables that occupy significant memory are being passed by reference, and are modified in place during the whole recursive procedure.

The following lemma shows how the recursive procedure to solve a $(T, \mathcal{B}, \alpha, \beta, \gamma)$-problem can be used to compute the rates of change of $Z^*(t)$ of a $(T, \mathcal{B})$-problem. Its proof follows from the observation that the rate of change of the solution with $t$ in (13) in Lemma 3.15 only depends on $\alpha$ and $\beta$, and that the reduction equations (14) in Lemma 3.16 never make $\alpha'$ or $\gamma'$ depend on $\beta$.

**Lemma 3.18** (Rates only). *Let $Z^*(t)$ be the solution of the $(T, \mathcal{B})$-problem, and let $\tilde{Z}^*(t)$ be the solution of the $(T, \mathcal{B}, \mathbf{1}, 0, \mathbf{1})$-problem. Then, $Z^*(t) = c_1 t + c_2$, and $\tilde{Z}^*(t) = c_1 t$ for some $c_1$ and $c_2$.*

We finally present the full algorithm to compute $Z'^*(t_i)$ and $\mathcal{L}'' * (t_i)$ from $T$ and $\mathcal{B}(t_i)$.

---

**Algorithm 3** ComputeRates (input: $T$ and $\mathcal{B}(t_i)$ output: $Z'^*(t_i)$ and $\mathcal{L}''(t_i)$)

---

1: $Z'^*(t_i)_j = 1$ for all $j \in \mathcal{B}(t_i)$
2: **for** each $(T_w, \mathcal{B}_w)$-problem induced by $\mathcal{B}(t_i)$ **do**
3:     Set $\tilde{T}_w$ to be $T_w$ pruned of all leaf nodes in $\mathcal{B}_w$, by repeatedly evoking Lemma 3.14
4:     ComputeRatesRec($\tilde{T}_w, j, \mathcal{B}_w, \mathbf{1}, \mathbf{0}, \mathbf{1}, \tilde{Z}'^*$)
5:     $Z'^*(t_i)_j = \tilde{Z}'^*{}_j$ for all $j \in V_w \backslash \mathcal{B}$
6: **end for**
7: Compute $\mathcal{L}''(t_i)$ from $Z'^*(t_i)$ using Lemma 3.12
8: **return** $Z'^*(t_i)$ and $\mathcal{L}''(t_i)$

---

The following theorem follows almost directly from Theorem 3.17.

**Theorem 3.19.** *Alg. 3 correctly computes $Z'^*(t_i)$ and $\mathcal{L}''(t_i)$ in $\mathcal{O}(q)$ steps, and uses $\mathcal{O}(q)$ memory.*

## 4 Reducing computation time in practice

Our numerical results are obtained for an improved version of Algorithm 1. We now explain the main idea behind this algorithm.

The bulk of the complexity of Alg. 1 comes from line 4, i.e., computing the rates $\{Z'^*(t_i)_j\}_{j \in \mathcal{V} \backslash \mathcal{B}(t_i)}$ from $\mathcal{B}(t_i)$ and $T$. For a fixed $j \in \mathcal{V} \backslash \mathcal{B}(t_i)$, and by Lemma 3.6, the rate $Z'^*(t_i)_j$, depends only on one particular $(T_w = (r_w, \mathcal{V}_w, \mathcal{E}_w), \mathcal{B}_w)$-problem induced by $\mathcal{B}(t_i)$. If exactly this same problem is induced by both $\mathcal{B}(t_i)$ and $\mathcal{B}(t_{i+1})$, which happens if the new nodes that become fixed in line 7 of round $i$ of Algorithm 1 are not in $\mathcal{V}_w \backslash \mathcal{B}_w$, then we can save computation time in round $i + 1$, by not recomputing any rates for $j \in \mathcal{V}_w \backslash \mathcal{B}_w$, and using for $Z'^*(t_{i+1})_j$ the value $Z'^*(t_i)_j$.

Furthermore, if only a few $\{Z_j'^*\}$ change from round $i$ to round $i + 1$, then we can also save computation time in computing $\mathcal{L}''$ from $Z'^*$ by subtracting from the sum in the right hand side of equation (11) the terms that depend on the previous, now changed, rates, and adding new terms that depend on the new rates.

Finally, if the rate $Z_j'^*$ does not change, then the value of $t < t_i$ at which $Z_j^*(t)$ might intersect $t - N_j$, and become fixed, given by $P_j$ in line 5, also does not change. (Note that this is not obvious from the formula for $P_r$ in line 5). If not all $\{P_r\}$ change from round $i$ to round $i + 1$, we can also save computation time in computing the maximum, and maximizers, in line 7 by storing $P$ in a maximum binary heap, and executing lines 5 and 7 by extracting all the maximal values from the top of the heap. Each time any $P_r$ changes, the heap needs to be updated.

## 5    Numerical results

Our algorithm to solve (3) exactly in a finite number of steps is of interest in itself. Still, it is interesting to compare it with other algorithms. In particular, we compare the convergence rate of our algorithm with two popular methods that solve (3) iteratively: the Alternating Direction Method of Multipliers (ADMM), and the Projected Gradient Descent (PGD) method. We apply the ADMM, and the PGD, to both the primal formulation (3), and the dual formulation (4). We implemented all the algorithms in C, and derived closed-form updates for ADMM and PG, see Appendix F. We ran all algorithms on a single core of an Intel Core i5 2.5GHz processor.

Figure 5-(left) compares different algorithms for a random Galton–Watson input tree truncated to have $q = 1000$ nodes, with the number of children of each node chosen uniformly within a fixed range, and for a random input $\hat{F} \in \mathbb{R}^q$, with entries chosen i.i.d. from a normal distribution. We observe the same behavior for all random instances that was tested. We gave ADMM and PGD an advantage by optimally tuning them for each individual problem-instance tested. In contrast, our algorithm requires no tuning, which is a clear advantage. At each iteration, the error is measured as $\max_j\{|M_j - M_j^*|\}$. Our algorithm is about $74\times$ faster than its closest competitor (PGD-primal) for $10^{-3}$ accuracy. In Figure 5-(right), we show the average run time of our algorithm versus the problem size, for random inputs of the same form. The scaling of our algorithm is (almost) linear, and much faster than our $\mathcal{O}(q^2 p)$, $p = 1$, theoretical bound.

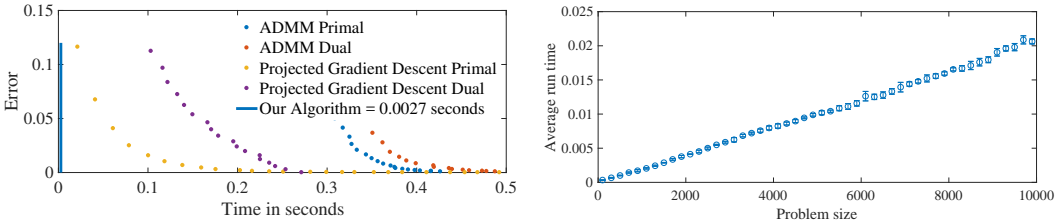

Figure 3: (Left) Time that the different algorithms take to solve our problem for trees of with 1000 nodes. (Right) Average run time of our algorithm for problems of different sizes. For each size, each point is averaged over 500 random problem instances.

Finally, we use our algorithm to exactly solve (2) by computing $\mathcal{C}(U)$ for all trees and a given input $\hat{F}$. Exactly solving (2) is very important for biology, since several relevant phylogenetic tree inference problems deal with trees of small sizes. We use an NVIDIA QUAD P5000 GPU to compute the cost of all possible trees with $q$ nodes in parallel, and return the tree with the smallest cost. Basically, we assign to each GPU virtual thread a unique tree, using Prufer sequences [31], and then have each thread compute the cost for its tree. For $q = 10$, we compute the cost of all 100 million trees in about 8 minutes, and for $q = 11$, we compute the cost of all 2.5 billion trees in slightly less than 2.5 hours.

Code to solve (3) using Alg. 1, with the improvements of Section 4, can be found in [32]. More results using our algorithm can be found in Appendix G.

## 6    Conclusions and future work

We propose a new direct algorithm that, for a given tree, computes how close the matrix of frequency of mutations per position is to satisfying the perfect phylogeny model. Our algorithm is faster than the state-of-the-art iterative methods for the same problem, even if we optimally tune them. We use the proposed algorithm to build a GPU-based phylogenetic tree inference engine for the trees of relevant biological sizes. Unlike existing algorithms, which only heuristically search a small part of the space of possible trees, our algorithm performs a complete search over all trees relatively fast. It is an open problem to find direct algorithms that can provably solve our problem in linear time on average, or even for a worst-case input.

**Acknowledgement:** This work was partially funded by NIH/1U01AI124302, NSF/IIS-1741129, and a NVIDIA hardware grant.

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
