[Supplementary Material]

# Appendix for "Efficient Projection onto the Perfect Phylogeny Model"

## A  Further illustrations

Figure 4: Four subtrees of $T$ induced by $\mathcal{B}(t)$, represented by the red squares. The root of $T_1$, $T_2$ and $T_4$ is node 1. The root of $T_3$ is node 4. All subtrees must have nodes associated to free variables (free nodes). Any subtree is uniquely identified by any free node in it. Within each subtree, any fixed node must be the root or a leaf.

## B  Proof of Theorem 3.1 in Section 3

We prove Theorem 3.1, by first proving the following very similar theorem.

**Theorem B.1.** *Problem* (3) *can be solved by solving*

$$\min_t t + \mathcal{L}(t), \tag{15}$$
$$\mathcal{L}(t) = \min_{Z \in \mathbb{R}^q} \frac{1}{2}\|(U^\top)^{-1}Z\|^2 \text{ subject to } Z + N \le t\mathbf{1}, \tag{16}$$

*where $N = U^\top \hat{F}$. In particular, if $t^*$ minimizes* (15), *$Z^*$ minimizes* (16) *for $t = t^*$, and $M^*$, $F^*$ minimize* (3), *then*

$$M^* = -U^{-1}(U^{-1})^\top Z^*, \quad F^* = -(U^{-1})^\top Z^*. \tag{17}$$

*Furthermore, $t^*$, $M^*$, $F^*$ and $Z^*$ are unique.*

*Proof of Theorem B.1.*  Problem (3) depends on the tree $T$ through the matrix of the ancestors, $U$. To see how Theorem B.1 implies Theorem 3.1, it is convenient to make this dependency more explicit. Any tree in $\mathcal{T}$, can be represented through a binary matrix $T$, where $T_{ij} = 1$ if and only if node $i$ is the closest ancestor of node $j$. Henceforth, let $\mathcal{T}$ denote the set of all such binary matrices. We need the following lemma, which we prove later in this section of the appendix.

**Lemma B.2.** *Consider an evolutionary tree and its matrices $T \in \mathcal{T}$ and $U \in \mathcal{U}$. We have*

$$U = (I - T)^{-1}. \tag{18}$$

Eq. (18) implies that $((U^{-1})^\top Z)_i = (Z - T^\top Z)_i = Z_i - Z_{\bar{i}}$, and that $U^{-1}((U^{-1})^\top Z)_i = Z_i - Z_{\bar{i}} - \sum_{r \in \partial i}(Z_r - Z_{\bar{r}})$, where $\partial i$ denotes the children of $i$ in $T$, $\bar{i}$ represents the closest ancestor of $i$ in $T$. We assume by convention that $Z_{\bar{i}} = 0$ when $i = r$ is the root of $T$. Furthermore, the definition of $U$ implies that $N_i = (U^\top \hat{F})_i = \sum_{j \in \Delta i} \hat{F}_j$, where $\Delta i$ denotes the ancestors of $j$. Thus,

$$\mathcal{L}(t) = \min_{Z \in \mathbb{R}^q} \frac{1}{2}\sum_{i \in \mathcal{V}}(Z_i - Z_{\bar{i}})^2 \text{ subject to} \tag{19}$$

$$Z_i \le t - \sum_{j \in \Delta i}\hat{F}_j \ , \forall i \in \mathcal{V},$$

$$M_i^* = -Z_i^* + Z_{\bar{i}}^* + \sum_{r \in \partial i}(Z_r^* - Z_{\bar{r}}^*) \text{ and} \tag{20}$$

$$F_i^* = -Z_i^* + Z_{\bar{i}}^*, \forall i \in \mathcal{V}.$$

$\square$

*Proof of Theorem 3.1.* Our proof is based on Moreau's decomposition [33]. Before we proceed with the proof, let us introduce a few concepts.

Given a convex, closed and proper function $g : \mathbb{R}^q \mapsto \mathbb{R}$, we define its proximal operator by the map $G : \mathbb{R}^q \mapsto \mathbb{R}^q$ such that

$$G(n) = \arg \min_{x \in \mathbb{R}^q} g(x) + \frac{1}{2}\|x - n\|^2, \tag{21}$$

where in our case $\| \cdot \|$ is the Euclidean norm. We define the Fenchel dual of $g$ as

$$g^*(x) = \sup_{s \in \mathbb{R}^q} \{x^\top s - g(s)\}, \tag{22}$$

and we denote the proximal operator of $g^*$ by $G^*$. Note that $G^*$ can be computed from definition (21) by replacing $g$ by $g^*$.

Moreau's decomposition identity states that

$$G(n) + G^*(n) = n. \tag{23}$$

We can now start the proof. Consider the following indicator function

$$g(\tilde{M}) = \begin{cases} 0, & \text{if } (U^{-1}\tilde{M}) \geq 0 \text{ and } \mathbf{1}^{\text{T}}(U^{-1}\tilde{M}) = 1, \\ +\infty, & \text{otherwise,} \end{cases} \tag{24}$$

where $\tilde{M} \in \mathbb{R}^q$, and consider its associated proximal operator $G$. Solving problem (3), i.e., finding a minimizer $M^*$, is equivalent to evaluating $U^{-1}G(\hat{F})$. Using Moreau's decomposition, we have

$$M^* = U^{-1}G(\hat{F}) = U^{-1}\hat{F} - U^{-1}G^*(\hat{F}). \tag{25}$$

We will show that $G^*(\hat{F}) = \hat{F} + (U^{-1})^\top Z^*$, where $Z^*$ is a minimizer of (5), which proves (6) and essentially completes the proof.

To compute $G^*$, we first need to compute

$$g^*(Y) = \sup_{\tilde{M}} \{Y^\top \tilde{M} - g(\tilde{M})\} \tag{26}$$

$$= \max_{\tilde{M}} Y^\top \tilde{M} \tag{27}$$

$$\text{subject to} \quad U^{-1}\tilde{M} \geq 0, \mathbf{1}^\top (U^{-1}\tilde{M}) = 1.$$

Making the change of variable $M = U^{-1}\tilde{M}$, the maximum in problem (27) can be re-written as

$$\max_{M} (U^\top Y)^\top M \tag{28}$$

$$\text{subject to} \quad M \geq 0, \mathbf{1}^\top M = 1.$$

It is immediate to see that the maximum in (28) is achieved if we set all components of $M$ equal to zero except the one corresponding to the largest component of the vector $U^\top Y$, which we should set to one. Therefore, we have

$$g^*(Y) = \max_i (U^\top Y)_i. \tag{29}$$

Now we can write

$$G^*(\hat{F}) = \arg \min_{Y \in \mathbb{R}^q} g^*(Y) + \frac{1}{2}\|Y - \hat{F}\|^2 \tag{30}$$

$$= \arg \min_{Y \in \mathbb{R}^q, t \in \mathbb{R}} t + \frac{1}{2}\|Y - \hat{F}\|^2 \tag{31}$$

$$\text{subject to} \quad U^\top Y \leq t.$$

Making the change of variable $Z = U^\top(Y - \hat{F})$, we can write $G^*(\hat{F})$ as

$$G^*(\hat{F}) = \hat{F} + (U^{-1})^\top Z^*, \text{ where} \tag{32}$$

$$(Z^*, t^*) = \arg\min_{Z \in \mathbb{R}^q, t \in \mathbb{R}} \ t + \frac{1}{2}\|(U^{-1})^\top Z\|^2 \tag{33}$$

$$\text{subject to} \quad Z + U^\top \hat{F} \leq t.$$

To see that $M^*$ and $F^*$ are unique, notice that problem (3) is a projection onto a convex set polytope, which always has a unique minimizer. Moureau's decomposition implies that $G^*(\hat{F})$ is unique, hence the minimizer $Y^*$ of (30) is unique. Thus, $Z^* = U^\top(Y^* - \hat{F})$ and $t^* = g^*(Y^*)$ are also unique. $\quad\square$

*Proof of Lemma B.2.* We assume that the tree has $q$ nodes. The matrix $T$ is such that $T_{v,v'} = 1$ if and only if $v$ is the closet ancestor of $v'$. Because of this, the $v$th column of $T^k$ has a one in row $v'$ if and only if $v'$ is an ancestor $v$ separated by $k$ generations. Thus, the $v$th column of $I + T + T^2 + \cdots + T^{q-1}$, contains a one in all the rows $v'$ such that $v'$ is an ancestor of $v$, or if $v = v'$. But this is the definition of the matrix $U$ associated to the tree $T$. Since no two mutants can be separated by more than $q - 1$ generations, $T^k = 0$ for all $k \geq q$. It follows that

$$U = I + T + T^2 + \cdots + T^{q-1} = \sum_{i=0}^{\infty} T^i = (I - T)^{-1}.$$

$\square$

# C  Proof of useful observations in Section 3.1

*Proof of Lemma 3.2.* The proof follows from the following generic fact, which we prove first. Let $g(W) = \min_{Z \in \mathbb{R}^q} f(Z, W)$. If $f$ is convex in $(Z, W)$, then $g$ is convex.

Indeed, let $\alpha \geq 0$ and $\alpha' = 1 - \alpha$. We get $\alpha g(W_1) + \alpha' g(W_2) = \min_{Z_1, Z_2} \alpha f(Z_1, W_1) + \alpha' f(Z_2, W_2) \geq \min_{Z_1, Z_2} f(\alpha Z_1 + \alpha' Z_2, \alpha W_1 + \alpha' W_2) = g(\alpha W_1 + \alpha' W_2)$.

To apply this result to our problem, let $f_1(Z)$ be the objective of (19) and let $f_2(Z, t, N)$ be a function (on the extended reals) such that $f_2 = 0$ if $(Z, t, N)$ satisfy the constraints in (19) and $+\infty$ otherwise. Now notice that $\mathcal{L}(t) = \min_Z f_1(Z) + f_2(Z, t)$, where $f_1 + f_2$ is convex in $(Z, t, N)$, since both $f_1$ and $f_2$ are convex in $(Z, t, N)$. Convexity implies that $\mathcal{L}$ is continuous in $N$ and $t$. It also implies that $\mathcal{L}'(t)$ is non increasing in $t$. $\quad\square$

*Proof of Lemma 3.3.* Continuity of $Z^*(t)$: The objective function in (19) is convex as a function of $Z$ and has unique minimum at $Z_i = 0, \forall i$. Hence, it is strictly convex. Due to strict convexity, if the objective takes values in a small interval, then $Z$ must be inside some small ball.

Since we know, by the remark following Lemma 3.2, that $\mathcal{L}$ is continuous as a function of $t$, if $t$ and $t'$ are close, then $\mathcal{L}(t)$ and $\mathcal{L}(t')$ must be close. Strict convexity then implies that $Z^*(t)$ and $Z^*(t')$ must be close. The same argument can be used to prove continuity with respect to $N$.

Continuity of $Z^*(t^*)$: Recall that $Z^*(t^*) = Z^*$, the solution of (3). $Z^*$ is a continuous function of $M^*$, which is the solution to (3), and thus is fully determined by $U$ and $\hat{F}$. Since, $\hat{F} = (U^\top)^{-1}N$, $\hat{F}$ is a continuous function of $N$, and it suffices to prove that $M^*$ is continuous in $\hat{F}$. Problem (3) finds the projection of $\hat{F}$ onto a convex polytope. Let $F^*$ be this projection. Since $F^*$ changes continuously with $\hat{F}$, $M^* = U^{-1}F^*$ also changes continuously with $\hat{F}$. $\quad\square$

*Proof of Lemma 3.4.* Since $Z^*(t)$ is continuous, if $Z^*(t)_i \neq t - N_i$ then $Z^*(t')_i \neq t' - N_i$ for $t'$ in some neighborhood of $t$. $\quad\square$

*Proof of Lemma 3.6.* First note that, by definition of $\mathcal{B}(t)$, we know the value of all variables in $\mathcal{B}(t)$. Hence, the unknowns in problem (19) are the variables in $\mathcal{V}\backslash\mathcal{B}(t)$, which can be partitioned into disjoint sets $\{\mathcal{V}_i\backslash\mathcal{B}(t)\}_{i=1}^k$.

Second notice that for each term in the objective (19) that involves not known variables, there is some subtree $T_i$ that contains both of its variables. It follows that, given $\mathcal{B}(t)$, problem (19) breaks into $k$ independent problems, the $i$th problem having as unknowns only the variables in $\mathcal{V}_i \setminus \mathcal{B}(t)$ and all terms in the objective where either $j$ or $\bar{j}$ are in $\mathcal{V}_i \setminus \mathcal{B}(t)$.

Obviously, if $j \in \mathcal{V}_w \cap \mathcal{B}(t)$, then, by definition, $Z^*(t)_j = c_1 t + c_2$, with $c_1 = 1$. To find the behavior of $Z^*(t)_j$ for $j \in \mathcal{V}_w \setminus \mathcal{B}(t)$, we need to solve 7. To solve (7), notice that the first-order optimality conditions for problem (19) imply that, if $j \in \mathcal{V} \setminus \mathcal{B}(t)$, then

$$Z_j = \frac{1}{|\partial j|} \sum_{r \in \partial j} Z_r, \tag{34}$$

where $\partial j$ denotes the neighbors of node $j$. We can further write

$$Z_j = \frac{1}{|\partial j|} \sum_{r \in \partial j \cap \mathcal{B}(t)} Z_r + \frac{1}{|\partial j|} \sum_{r \in \partial j \setminus \mathcal{B}(t)},$$

$$Z_r = \frac{1}{|\partial j|} \sum_{r \in \partial j \cap \mathcal{B}(t)} (t - N_r) + \frac{1}{|\partial j|} \sum_{r \in \partial j \setminus \mathcal{B}(t)} Z_r. \tag{35}$$

It follows that $Z_j = c_1 t + c_2$, for some $c_1$ and $c_2$ that depend on $T$, $N$ and $\mathcal{B}$. If we solve for $Z_j$ by recursively applying (35), it is immediate to see that $c_1 \geq 0$.

To see that $c_1 \leq 1$, we study how $Z_j$, defined by (35), depends on $t$ algebraically. To do so, we treat $t$ as a variable. The study of this algebraic dependency in the proof should not be confused with $t$ being fixed in the statement of the theorem.

Define $\rho = |\partial i \cup \mathcal{B}(t)| / |\partial j|$, and notice that

$$\max_j \{Z_j\} \leq \rho t + (1 - \rho) \max_j \{Z_j\} + C, \tag{36}$$

in which $C$ is some constant. Recursively applying the above inequality we get

$$\max_j \{Z_j\} \leq t + C', \tag{37}$$

in which $C'$ is some constant. This shows that no $Z_j$ can grow with $t$ faster than $1 \times t$ and hence $c_1 \leq 1$. □

*Proof of Lemma 3.7.* Lemma 3.6 implies that, for any $j$, $Z_j^*(t)$ depends linearly on $t$. The particular linear dependency, depends on $\mathcal{B}(t)$, which is piecewise constant by Lemma 3.4. Therefore, $Z_j^*(t)$ is a continuous piecewise linear function of $t$. This in turn implies that $\mathcal{L}'(t)$ is a continuous piecewise linear function of $t$, since it is the derivative of the continuous piecewise quadratic $\mathcal{L}(t) = (1/2) \sum_{i \in \mathcal{V}} (Z^*(t)_i - Z^*(t)_{\bar{i}})^2$. Finally, since the particular linear dependency of $Z^*$, depends on $\mathcal{B}(t)$, it follows that $Z^*(t)$ and $\mathcal{L}'(t)$ change linear segment if and only if $\mathcal{B}(t)$ changes. □

*Proof of Lemma 3.8.* Let us assume that there exists $t < t'$ for which $\mathcal{B}(t) \subset \mathcal{B}(t')$. We can assume without loss of generality that $t$ is sufficiently close to $t'$ such that $\mathcal{B}(s)$ is constant for $s \in [t, t')$. Let $j$ be such that $j \in B(t')$ but $j \notin B(t)$. This means that $Z_j^*(s) < s - N_j$ for all $s \in [t, t')$ and that $Z_j^*(t') = t' - N_j$. Since by Lemma 3.6, $Z_j^*(s) = c_1 s + c_2$, for some constants $c_1$ and $c_2$, the only way that $Z_j^*(s)$ can intersect $s - N_j$ at $s = t'$ is for $c_1 > 1$, which is a contradiction.

If $\mathcal{B}(t)$ decreases as $t$ increase, and given that the largest that $\mathcal{B}(t)$ can be is $\{1, \ldots, q\}$, it follows that $\mathcal{B}(t)$ can only take $q + 1$ different configurations. One configuration per size of $\mathcal{B}(t)$, from $q$ to 0. □

*Proof of Lemma 3.9.* Lemma 3.8 implies that $\mathcal{B}(t)$ changes at most $q + 1$ times. Lemma 3.7 then implies that $Z^*(t)$ and $\mathcal{L}'(t)$ have less than $q + 1$ different linear segments. □

# D   Proofs of the properties of the algorithm in Section 3.2

*Proof of Theorem 3.10.* **Run-time:** Recall that $Z^*, Z'^* \in \mathbb{R}^q$ and that $\mathcal{L}' \in \mathbb{R}$. Line 1 is done in $\mathcal{O}(q)$ steps by doing a DFS on $T$. Here, we assume that $T$ is represented as a linked list. Specifically,

starting from the root, we keep a variable $x$ where we accumulate the values of $\hat{F}_j$ visited from the root to the current node being explored in $T$ as we move down the tree. As we move up the tree, we subtract values of the nodes $\hat{F}_j$ from $x$. Then, at each node $i$ visited by the DFS, we can read from $x$ the value $N_i$. Line 2 takes $\mathcal{O}(q)$ steps to finish. The procedure ComputeRates takes $\mathcal{O}(q)$ steps to finish, which we prove in Theorem 3.19. All of the other lines inside the for-loop are manipulations that take at most $\mathcal{O}(q)$ steps. Lines 13 and 12 take $\mathcal{O}(q)$ steps. From (6), the complexity to compute $F^*$ is $\mathcal{O}(q)$, and the complexity to compute $M^*$ is $\mathcal{O}(\sum_{i \in \mathcal{V}} |\partial i|) = \mathcal{O}(|\mathcal{E}|) = \mathcal{O}(q)$.

**Memory:** The DFS in line 1 only requires $\mathcal{O}(q)$ memory. Throughout the algorithm, we only need to keep the two most recent values of $t_i$, $\mathcal{B}(t_i)$, $Z^*(t_i)$, $Z'^*(t_i)$, $\mathcal{L}'(t_i)$ and $\mathcal{L}''(t_i)$. This takes $\mathcal{O}(q)$ memory. The procedure ComputeRates takes $\mathcal{O}(q)$ memory, which we prove in Theorem 3.19. □

*Proof of Theorem 3.11.* The proof of Theorem 3.11 amounts to checking that, at every step of Algorithm 1, the quantities computed, e.g., the paths $\{Z^*(t)\}$ and $\{\mathcal{L}'(t)\}$, are correct.

Lemmas 3.7 and 3.9 prove that $Z^*(t)$ and $\mathcal{L}'(t)$ are piecewise linear and continuous with at most $q$ changes in linear segment. Hence, the paths $\{Z^*(t)\}$ and $\{\mathcal{L}'(t)\}$ are fully specified by their value at $\{t_i\}_{i=1}^{k}$, and $k \leq q$.

Lemma 3.7 proves that these critical values are determined as the instants, at which $\mathcal{B}(t)$ changes. Furthermore, Lemma 3.8 proves that, as $t$ decreases, variables are only added to $\mathcal{B}(t)$. Hence, to find $\{t_i\}$ and $\{\mathcal{B}(t_i)\}$, we only need to find the times and components at which, as $t$ decreases, $Z^*(t)_r$ goes from $Z^*(t)_r < t - N_r$ to $Z^*(t)_r = t - N_r$. Also, since $\mathcal{B}$ can have at most $q$ variables, the for-loop in line 3 being bounded to the range 1-$q$, does not prevent the algorithm from finding any critical value.

Theorem 3.19 tells us that we can compute $Z'^*(t_i)$ from $\mathcal{B}(t_i)$ and $T$. Since we have already proved that the path $\{Z^*(t)\}$ is piecewise linear and continuous, we can compute $t_{i+1}$, and the variables that become fixed, by solving (10) for $t$ for each $r \notin \mathcal{B}(t_i)$, and choosing for $t_{i+1}$ the largest such $t$, and choosing for the new fixed variables, i.e., $\mathcal{B}(t_{i+1}) - \mathcal{B}(t_i)$, the components $r$ for which the solution of (10) is $t_{i+1}$.

Since we have already proved that that $Z^*(t)$ and $\mathcal{L}'(t)$ are piecewise linear and constant, we can compute $Z^*(t_{i+1})$ and $\mathcal{L}'(t_{i+1})$ from $Z^*(t_i)$, $\mathcal{L}'(t_i)$, $Z'^*(t_i)$ and $\mathcal{L}''(t_i)$ using (9).

Lemma 3.2 proves that $\mathcal{L}'(t)$ decreases with $t$, and Theorem 3.1 proves that $t^*$ is unique. Hence, as $t$ decreases, there is a single $t$ at which $\mathcal{L}'(t)$ goes from $> -1$ to $< -1$. Since we have already proved that we correctly, and sequentially, compute $\mathcal{L}'(t_i)$, $\mathcal{L}''(t_i)$, and that $\mathcal{L}'(t)$ is piecewise linear and constant, we can stop computing critical values whenever we can determine that $\mathcal{L}'(t) = \mathcal{L}'(t_k) + (t - t_k)\mathcal{L}''(t_k)$ will cross the value $-1$, where $t_k$ is the latest computed critical value. This is the case when $\mathcal{L}'(t_k) > -1$ and $\mathcal{L}'(t_{k+1}) < -1$, or when $\mathcal{L}'(t_k) > -1$ and $t_k$ is the last possible critical value, which happens when $|\mathcal{B}(t_i)| = q$. From this last critical value, $t_k$, we can then find $t^*$ and $Z^*$ by solving $-1 = \mathcal{L}'(t_k) + (t^* - t_k)\mathcal{L}''(t_k)$ and $Z^* = Z^*(t_k) + (t^* - t_k)Z'^*(t_k)$. Finally, once we have $Z^*$, we can use (6) in Theorem 3.1 to find $M^*$ and $F^*$. □

## E  Proofs for computing the rates in Section 3.3

*Proof of Lemma 3.12.* Let $t \in (t_{i+1}, t_i)$. We have,

$$\mathcal{L}'(t) = \frac{\mathrm{d}}{\mathrm{d}t} \frac{1}{2} \sum_{j \in \mathcal{V}} (Z^*(t)_j - Z^*(t)_{\bar{j}})^2 =$$

$$\sum_{i \in \mathcal{V}} (Z^*(t)_j - Z^*(t)_{\bar{j}})(Z^{*'}(t)_j - Z^{*'}(t)_{\bar{j}}). \tag{38}$$

Taking another derivative, and recalling that $Z''^*(t) = 0$ for $t \in (t_{i+1}, t_i)$, we get

$$\mathcal{L}''(t) = \sum_{j \in \mathcal{V}} (Z^{*'}(t)_j - Z^{*'}(t)_{\bar{j}})^2, \tag{39}$$

and the lemma follows by taking the limit $t \uparrow t_i$. □

*Proof of Lemma 3.14.* The $(T, \mathcal{B}, \alpha, \beta, \gamma)$-problem is unconstrained and convex, hence we can solve it by taking derivatives of the objective with respect to the free variables, and setting them to zero. Let us call the objective function $F(Z)$. If $j \in \mathcal{V}\backslash\mathcal{B}$ is a leaf, then $\frac{\mathrm{d}F}{\mathrm{d}Z_j} = 0$ implies that $Z_j^* = Z_{\bar{j}}^*$. We now prove the second part of the lemma. Let $\tilde{F}(Z)$ be the objective of the modified problem. Clearly, $\frac{\mathrm{d}F}{\mathrm{d}Z_i} = \frac{\mathrm{d}\tilde{F}}{\mathrm{d}Z_i}$ for all $i \in \tilde{T}\backslash\bar{j}$. Let $C$ be the children of $\bar{j}$ in $T$ and $\tilde{C}$ be the children of $\bar{j}$ in $\tilde{T}$. We have $\tilde{C} = C\backslash j$. Furthermore, $\frac{\mathrm{d}\tilde{F}}{\mathrm{d}Z_j} = 0$ is equivalent to $\gamma_j(Z_j - Z_{\bar{j}}) + \sum_{s \in \tilde{C}} \gamma_s(Z_j - Z_s) = 0$, and $\frac{\mathrm{d}F}{\mathrm{d}Z_j} = 0$ is equivalent to $\gamma_j(Z_j - Z_{\bar{j}}) + \sum_{s \in C} \gamma_s(Z_j - Z_s) = 0$. However, we have already proved that the optimal solution for the original problem has $Z_j^* = Z_{\bar{j}}^*$. Hence, this condition can be replaced in $\frac{\mathrm{d}F}{\mathrm{d}Z_j}$, which becomes $\gamma_j(Z_j - Z_{\bar{j}}) + \sum_{s \in \tilde{C}} \gamma_s(Z_j - Z_s) = 0$. Therefore, the two problems have the same optimality conditions, which implies that $Z_i^* = \tilde{Z}_i^*$, for all $i \in \tilde{\mathcal{V}}$. $\qquad\square$

*Proof of Lemma 3.15.* The proof follows directly from the first order optimality conditions, a linear equation that we solve for $Z_1^*$. $\qquad\square$

*Proof of Lemma 3.16.* The first order optimality conditions for both problems are a system of linear equations, one equation per free node in each problem. All the equations associated to the ancestral nodes of $j$ are the same for both problems. The equation associated to variable $j$ in the $(T, \mathcal{B}, \alpha, \beta, \gamma)$-problem is

$$\gamma_j(Z_{\bar{j}} - Z_j) + \sum_{i=1}^{r} \gamma_i(Z_i - Z_j) = 0, \tag{40}$$

which implies that

$$Z_j = \frac{\gamma_j Z_{\bar{j}} + \sum_{i=1}^{r} \gamma_i Z_i}{\gamma_j + \sum_{i=1}^{r} \gamma_i}. \tag{41}$$

The equation associated to the variable $\bar{j}$ in the $(T, \mathcal{B}, \alpha, \beta, \gamma)$-problem is

$$F(Z, \alpha, \beta, \gamma) + \gamma_j(Z_j - Z_{\bar{j}}) = 0, \tag{42}$$

where $F(Z)$ is a linear function of $Z$ determined by the tree structure and parameters associated to the ancestral edges and nodes of $\bar{j}$. The equation associated to the variable $\bar{j}$ in the $(\tilde{T}, \tilde{\mathcal{B}}, \tilde{\alpha}, \tilde{\beta}, \tilde{\gamma})$-problem is

$$F(\tilde{Z}, \tilde{\alpha}, \tilde{\beta}, \tilde{\gamma}) + \tilde{\gamma}_j(\tilde{\alpha}_j t + \tilde{\beta}_j - \tilde{Z}_{\bar{j}}) = 0, \tag{43}$$

for the same function $F$ as in (42). Note that the components of $\tilde{\alpha}$, $\tilde{\beta}$ and $\tilde{\gamma}$ associated to the ancestral edges and nodes of $\bar{j}$ are the same as in $\alpha$, $\beta$ and $\gamma$. Hence, $F(\tilde{Z}, \tilde{\alpha}, \tilde{\beta}, \tilde{\gamma}) = F(\tilde{Z}, \alpha, \beta, \gamma)$.

By replacing (41) into (42), one can easily check the following. Equations (42) and (43), as linear equations on $Z$ and $\tilde{Z}$ respectively, have the same coefficients if (14) holds. Hence, if (14) holds, the solution to the linear system associated to the optimality conditions in both problem gives the same optimal value for all variables ancestral to $\bar{j}$ and including $\bar{j}$. $\qquad\square$

*Proof of Theorem 3.17.* Although $T$ changes during the execution of the algorithm, in the proof we let $T = (r, \mathcal{V}, \mathcal{E})$ be the tree, passed to the algorithm at the zeroth level of the recursion. Recall that $|\mathcal{V}| = q$ and $\mathcal{E} = q - 1$.

**Correctness:** The correctness of the algorithm follows directly from Lemmas 3.14, 3.15, and 3.16 and the explanation following these lemmas.

**Run-time:** It is convenient to think of the complexity of the algorithm by assuming that it is running on a machine with a single instruction pointer that jumps from line to line in Algorithm 2. With this in mind, for example, the recursive call in line 6 simply makes the instruction pointer jump from line 6 to line 1. The run-time of the algorithm is bounded by the sum of the time spent in each line in Algorithm 2, throughout its entire execution. Each basic step costs one unit of time. Each node in $\mathcal{V}$ is only chosen as $j$ at most once, throughout the entire execution of the algorithm. Hence, line 1 is executed at most $q$ times, and thus any line is executed at most $q$ times, at most once for each possible choice for $j$.

Assuming that we have $T.b$ updated, $j$ in line 1 can be executed in $\mathcal{O}(1)$ time, by reading the first element of the linked list $T.b$. Lines 2 and 6 also take $\mathcal{O}(1)$ time. Here, we are thinking of the cost of line 6 as simply the cost to make the instruction pointer jump from line 1 to line 6, not the cost to fully completing the call to *ComputeRatesRec* on the modified problem. The modification made to the $(T, \mathcal{B}, \alpha, \beta, \gamma)$-problem by lines 5 and 7, is related to the addition, or removal, of at most degree$(j)$ nodes, where degree$(j)$ is the degree of $j$ in $T$. Hence, they can be executed in $\mathcal{O}(\text{degree}(j))$ steps. Finally, lines 3 and 8 require solving a star-shaped problem with $\mathcal{O}(\text{degree}(j))$ variables, and thus take $\mathcal{O}(\text{degree}j)$, which can be observed by inspecting (13).

Therefore, the run-time of the algorithm is bounded by $\mathcal{O}(\sum_j \text{degree}(j)) = \mathcal{O}(q)$.

To see that it is not expensive to keep $T$ updated, notice that, if $T$ changes, then either $T.b$ loses $j$ (line 5) or has $j$ reinserted (line 7), both of which can be done in $\mathcal{O}(1)$ steps. Hence, we can keep $T.b$ updated with only $\mathcal{O}(1)$ effort each time we run line 5 and line 7. Throughout the execution of the algorithm, the tree $T$ either shrinks by loosing nodes that are children of the same parent (line 5), or $T$ grows by regaining nodes that are all siblings (line 7). Hence, the linked list $T.a$ can be kept updated with only $\mathcal{O}(1)$ effort each time we run line 5 and line 7. Across the whole execution of the algorithm, $T.a$ and $T.b$ can be kept updated with $\mathcal{O}(\sum_j \text{degree}(j)) = \mathcal{O}(q)$ effort.

**Memory:** All the variables with a size that depend on $q$ are passed by reference in each call of *ComputeRatesRec*, namely, $Y, T, \mathcal{B}, \alpha, \beta$ and $\gamma$. Hence, we only need to allocate memory for them once, at the zeroth level of the recursion. All these variables take $\mathcal{O}(q)$ memory to store. $\qquad\square$

*Proof of Lemma 3.18.* From Definition 3.13, we know that the $(T, \mathcal{B}, \mathbf{1}, -N, \mathbf{1})$-problem and the $(T, \mathcal{B})$-problem are the same. Hence, it is enough to prove that the solutions of (i) any $(T, \mathcal{B}, \alpha, \beta, \gamma)$-problem and of (ii) the $(T, \mathcal{B}, \alpha, 0, \gamma)$-problem change at the same rate as a function of $t$.

We have already seen that the $(T, \mathcal{B}, \alpha, \beta, \gamma)$-problem can be solved by recursively invoking Lemma 3.16 until we arrive at problems that are small enough to be solved via Lemma 3.15.

We now make two observations. First, while recursing, Lemma 3.16 always transform a $(\tilde{T}, \tilde{\mathcal{B}}, \tilde{\alpha}, \tilde{\beta}, \tilde{\gamma})$-problem into a smaller problem $(\tilde{\tilde{T}}, \tilde{\tilde{\mathcal{B}}}, \tilde{\tilde{\alpha}}, \tilde{\tilde{\beta}}, \tilde{\tilde{\gamma}})$-problem where, by (14), $\tilde{\tilde{\gamma}}$ and $\tilde{\tilde{\alpha}}$ only depend on $\tilde{\alpha}$ and $\tilde{\gamma}$ but not on $\tilde{\beta}$.

Second, while recursing, and each time Lemma 3.15 is invoked to compute an explicit value for some component of the solution via solving some star-shaped $(\tilde{\tilde{T}}, \tilde{\tilde{\mathcal{B}}}, \tilde{\tilde{\alpha}}, \tilde{\tilde{\beta}}, \tilde{\tilde{\gamma}})$-problem, the rate of change of this component with $t$, is a function of $\tilde{\tilde{\alpha}}$ and $\tilde{\tilde{\gamma}}$ only. We can see this from (13).

Hence, the rate of change with $t$ of the solution of the $(T, \mathcal{B}, \alpha, \beta, \gamma)$-problem does not depend on $\beta$. So we can assume $\beta = 0$. $\qquad\square$

*Proof of Theorem 3.19.* **Correctness:** The correctness of Algorithm 3 follows from the correctness of Algorithm 2.

**Run-time and memory:** We can prune each $T_w$ in $\mathcal{O}(|T_w|)$ steps and $\mathcal{O}(1)$ memory using DFS. In particular, once we reach a leaf of $T_w$ that is free, i.e., not in $\mathcal{B}_w$, and as DFS travels back up the tree, we can prune from $T_w$ all the nodes that are free. By Theorem 3.17, the number of steps and memory needed to completely finish line 4 is $\mathcal{O}(|T_w|)$. The same is true to complete line 5. Hence, the number of steps and memory required to execute the for-loop is $\mathcal{O}(\sum_w |T_w|) = \mathcal{O}(|T|) = \mathcal{O}(q)$. Finally, by Theorem 3.12, $\mathcal{L}''$ can be computed from $Z'^*$ in $\mathcal{O}(q)$ steps using $\mathcal{O}(1)$ memory. $\qquad\square$

# F Details of the ADMM and the PGD algorithms in Section 5

Here we explain the details of our implementations of the Alternating Direction Method of Multipliers (ADMM) and the Projected Gradient Descent (PGD) methods, applied to our problem.

### F.1 ADMM

#### F.1.1 ADMM for the primal problem

We start by putting our initial optimization problem (3) into the following equivalent form:

$$\min_{M \in \mathbb{R}^q} \left\{ f(M) = \frac{1}{2} \|F - UM\|^2 \right\} + g(M), \tag{44}$$

where $g(M)$ is the indicator function imposing the constraints on $M$:

$$g(M) := \begin{cases} 0, & M \geq 0, M^\top \mathbf{1} = \mathbf{1}, \\ +\infty, & \text{otherwise.} \end{cases} \tag{45}$$

In this formulation, our target function is a sum of two terms. We now proceed with the standard ADMM procedure, utilizing the splitting $f, g$. Our ADMM scheme iterates on the following variables $M, M_1, M_2, u_1, u_2, \in \mathbb{R}^q$. $M_1$ and $M_2$ are primal variables, $M$ is a consensus variable, and $u_1$ and $u_2$ are dual variables. It has tunning parameters $\alpha, \rho \in \mathbb{R}$.

First, we evaluate the proximal map associated with the first term

$$M_1 \leftarrow \arg\min_{S \in \mathbb{R}^q} \frac{1}{2} \|F - US\|^2 + \frac{\rho}{2} \|S - M + u_1\|^2, \tag{46}$$

where $S$ is a dummy variable. This map can be evaluated in closed form,

$$M_1 = (\rho I + U^\top U)^{-1} (\rho M - \rho u_1 + U^\top F). \tag{47}$$

Second, we evaluate the proximal map associated with the second term

$$M_2 \leftarrow \arg\min_{S \in \mathbb{R}^q} g(S) + \frac{\rho}{2} \|S - M + u_2\|^2, \tag{48}$$

where $S$ is again a dummy variable. This map is precisely the projection onto the simplex, which has been extensively studied in the literature; there are many fast algorithms that solve this problem exactly. We implemented the algorithm proposed in [16].

Lastly, we perform the rest of the standard ADMM updates:

$$\begin{aligned} M &\leftarrow \frac{1}{2}(M_1 + u_1 + M_2 + u_2), \\ u_1 &\leftarrow u_1 + \alpha(M_1 - M), \\ u_2 &\leftarrow u_2 + \alpha(M_2 - M). \end{aligned} \tag{49}$$

We repeat the above steps until a satisfactory precision is reached, and read off the final solution from the variable $M$.

#### F.1.2 ADMM for the dual problem

We now apply ADMM to the dual problem (4). We start by incorporating the constraints into the target function to rewrite (4) as

$$\min_{Z,t} \{ f(t) = t \} + \left\{ h(Z) = \frac{1}{2} \|(U^\top)^{-1} Z\|^2 \right\} + g(t, Z), \tag{50}$$

where

$$g(t, Z) := \begin{cases} 0, & t\mathbf{1} - Z \geq N, \\ +\infty, & \text{otherwise,} \end{cases} \tag{51}$$

is the indicator function imposing the constraints on $t, Z$. ADMM now splits the problem into three parts, each associated to one of the functions $f, g$ and $h$.

Our ADMM scheme will iterate on the following variables $Z, X_Z, X_{gZ}, u_Z, u_{gZ} \in \mathbb{R}^q$, and $t, X_t, X_{gt}, u_t, u_{gt} \in \mathbb{R}$. The variables $X_Z, X_{gZ}, X_t, X_{gt}$ are primal variables, $t, Z$ are consensus variables, and $u_Z, u_{gZ}, u_t, u_{gt}$ are dual variables. It has tunning parameters $\alpha, \rho \in \mathbb{R}$.

First, we evaluate the proximal map for the first term

$$X_Z \leftarrow \arg\min_{S \in \mathbb{R}^q} \frac{1}{2}\|(U^\top)^{-1}S\|^2 + \frac{\rho}{2}\|S - Z + u_Z\|^2, \qquad (52)$$

where $S$ is a dummy variable. This map can be evaluated using an closed form formula:

$$X_Z = (\rho I + U^{-1}(U^{-1})^\top)^{-1}\rho(Z - u_Z). \qquad (53)$$

Next, we evaluate the proximal map for the second term

$$X_t \leftarrow \arg\min_{S \in \mathbb{R}} S + \frac{\rho}{2}(S - t + u_t)^2, \qquad (54)$$

where $S$ is a dummy variable. Again, this can be solved straightforwardly:

$$X_t = \frac{\rho t - \rho u_t - 1}{\rho}. \qquad (55)$$

We then evaluate the proximal map for the third term, which involves the constraints

$$(X_{gZ}, X_{gt}) \leftarrow \arg\min_{S \in \mathbb{R}^q, S_t \in \mathbb{R}} g(S, S_t) + \frac{\rho}{2}\|(S, S_t)$$
$$- (Z - u_{gZ}, t - u_{gt})\|^2, \qquad (56)$$

where $S, S_t$ are dummy variables. This problem is a projection onto the polyhedron defined by the constraints, $t\mathbf{1} - Z \geq N$, in $\mathbb{R}^{q+1}$. We developed an algorithm that solves this problem exactly in $\mathcal{O}(q \log q)$ steps. This is discussed in Section F.3.

What is left to be done is the following part of the ADMM:

$$
\begin{aligned}
Z &\leftarrow \frac{1}{2}(X_Z + u_Z + X_{gZ} + u_{gZ}), \\
u_Z &\leftarrow u_Z + \alpha(X_Z - Z), \\
u_{gZ} &\leftarrow u_{gZ} + \alpha(X_{gZ} - Z), \\
t &\leftarrow \frac{1}{2}(X_t + u_t + X_{gt} + u_{gt}), \\
u_t &\leftarrow u_t + \alpha(X_t - t), \\
u_{gt} &\leftarrow u_{gt} + \alpha(X_{gt} - t).
\end{aligned}
\qquad (57)
$$

We repeat the above steps until a satisfactory precision is reached, and read off the final solution from the variables $t$ and $Z$.

## F.2 PGD

### F.2.1 PGD for the primal problem

Implementing PGD is rather straightforward. For the initial problem (3), we simply do the following update:

$$M \leftarrow \text{Proj-onto-Simplex}(M + \alpha U^\top(F - UM)), \qquad (58)$$

where Proj-onto-Simplex() refers to projection onto the simplex, for which we implemented the algorithm proposed in [16]. $\alpha \in \mathbb{R}$ is the step size, a tuning parameter. We perform this update repeatedly until a satisfactory precision is reached.

### F.2.2 PGD for the dual problem

For the dual problem (4), the updates we need are

$$
\begin{aligned}
Z &\leftarrow Z - \alpha U^{-1}(U^{-1})^\top Z, \\
t &\leftarrow t - \alpha, \\
(Z, t) &\leftarrow \text{Proj-onto-Polyhedron}((Z, t)),
\end{aligned}
\qquad (59)
$$

where Proj-onto-Polyhedron() refers to projection onto the polyhedron defined by $t\mathbf{1} - Z \geq N$ in $\mathbb{R}^{q+1}$, while $\alpha \in \mathbb{R}$ is the step size. This is explicitly explained in F.3. Again, we perform these updates repeatedly until a satisfactory precision is reached, and tune the parameters to achieve the best possible performance.

### F.3 Projection onto the polyhedron $t\mathbf{1} - Z \geq N$

We would like to solve the following optimization problem:

$$\underset{Z \in \mathbb{R}^q, t \in \mathbb{R}}{\arg\min} \frac{1}{2} \|(Z, t) - (A, B)\|^2, \tag{60}$$

$$\text{subject to} \quad t\mathbf{1} - Z \geq N, \tag{61}$$

which is the problem of projection onto the polyhedron $t\mathbf{1} - Z \geq N$ in $\mathbb{R}^{q+1}$. The Lagrangian of this optimization problem is

$$\mathcal{L} = \frac{1}{2} \|(Z, t) - (A, B)\|^2 + \lambda^\top (Z + N - t\mathbf{1}), \tag{62}$$

where $\lambda \in \mathbb{R}^q$ is the Lagrange multiplier. We solve problem (60) by solving the dual problem $\max_{\lambda \geq 0} \min_{Z, t} \mathcal{L}$.

We first solve the minimization over variables $Z$ and $t$. It is straightforward to find the closed form solutions:

$$Z^* = A - \lambda, \qquad t^* = B + \mathbf{1}^\top \lambda. \tag{63}$$

Using these expressions, we can rewrite the Lagrangian as

$$\mathcal{L} = -\frac{1}{2} \lambda^\top (I + \mathbf{1}\mathbf{1}^\top) \lambda + R^\top \lambda, \tag{64}$$

where $R = A + N - B\mathbf{1}$.

Now our goal becomes solving the following optimization problem:

$$\arg\min \frac{1}{2} \lambda^\top (I + \mathbf{1}\mathbf{1}^\top) \lambda - R^\top \lambda, \tag{65}$$

$$\text{subject to} \quad \lambda \geq 0. \tag{66}$$

The KKT conditions for (65) are

$$\lambda_i + \mathbf{1}^\top \lambda - R_i - s_i = 0, \ \ \lambda_i \geq 0, \ \ s_i \geq 0, \ \ \lambda_i s_i = 0, i = 1, .., q, \tag{67}$$

where $s_i$ are Lagrange multipliers associated with the constraint $\lambda \geq 0$.

We proceed with sorting the vector $R$ first, and maintain a map $f : \{1, 2, ..., q\} \rightarrow \{1, 2, ..., q\}$ that maps the sorted indices back to the unsorted indices of $R$. Let us call the sorted $R$ by $\tilde{R}$. Then, from the above KKT conditions, it is straightforward to derive the following expression for $\lambda_i$:

$$\lambda_i = \begin{cases} \tilde{R}_i - \mathbf{1}^\top \lambda, \ i \geq \tau, \\ 0, \ i < \tau, \end{cases} \quad i = 1, 2, ..., q \tag{68}$$

where

$$\tau = \min\{i \mid \tilde{R}_i - \mathbf{1}^\top \lambda \geq 0\}. \tag{69}$$

Then it follows that

$$\mathbf{1}^\top \lambda = \sum_{i=\tau}^{q} (\tilde{R}_i - \mathbf{1}^\top \lambda) = \frac{1}{2 + q - \tau} \sum_{i=\tau}^{q} \tilde{R}_i, \tag{70}$$

and hence we have that

$$c(\tau) := \tilde{R}_\tau - \mathbf{1}^\top \lambda = \tilde{R}_\tau - \frac{1}{2 + q - i} \sum_{j=\tau}^{q} \tilde{R}_j. \tag{71}$$

According to (69), to find $\tau$, we only need to find the smallest value of $i$ that makes $c(i)$ non negative. That is, $\tau = \min\{i \mid c(i) \geq 0\}$.

Therefore, by sorting the components of $R$ from small to large, and checking $c(i)$ for each component, from large $i$ to small $i$, we can obtain the desired index $\tau$. Combining equations (68) and (70) with $\tau$, we find a solution that equals $\lambda^*$, the solution to problem (65), apart from a permutation of its components. We then use our index map $f$ to undo the sorting of the components introduced by sorting $R$.

Finally, by plugging $\lambda^*$ back into equation (63), we obtain the desired solution to our problem (60). The whole projection procedure can be done in $\mathcal{O}(q \log q)$, the slowest step being the sorting of $R$.

# G  More results using our algorithm

In this section, we use our fast projection algorithm to infer phylogenetic trees from frequency of mutation data.

The idea is simple. We scan all possible trees, and, for each tree $T$, we project $\hat{F}$ into a PPM for this $T$ using our fast projection algorithm. This gives us a projected $F$ and $M$ such that $F = UM$, the columns of $M$ are in the probability simplex, and $\|\hat{F} - F\|$ is small. Then, we return the tree whose projection yields the smallest $\|\hat{F} - F\|$. Since all of these projections can be done in parallel, we assign the projection for different subsets of the set of all possible trees to different GPU cores. Since we are performing an exhaustive search over all possible trees, we can only infer small trees. As such, when dealing with real-size data, similar to several existing tools, we first cluster the rows of $\hat{F}$, and produce an "effective" $\hat{F}$ with a small number of rows. We infer a tree on this reduced input. Each node in our tree is thus associated with multiple mutated positions in the genome, and multiple mutants, depending on the clustering. We cluster the rows of $\hat{F}$ using $k$-means, just like in [6]. We decide on the numbers of clusters, and hence tree size, based on the same BIC procedure as in [6]. It is possible that other pre-clustering, and tree-size-selection strategies, yield better results. We call the resulting tool EXACT.

We note that it is not our goal to show that the PPM is adequate to extract phylogenetic trees from data. This adequacy, and its limits, are well documented in well-cited biology papers. Indeed, several papers provide open-source tools based on the PPM, and show their tools' good performance on data containing the frequencies of mutation per position in different samples, $\hat{F}$ in our paper. A few tools are PhyloSub [5], AncesTree [3], CITUP [6], PhyloWGS [14], Canopy [34], SPRUCE [35], rec-BTP [36], and LICHeE [7]. These papers also discuss the limitations of the PPM regarding inferring evolutionary trees, and others propose extensions to the PPM to capture more complex phenomena, see e.g., [37].

It is important to further distinguish the focus of our paper from the focus of the papers cited in the paragraph above. In this paper, we start from the fact that the PPM is already being used to infer trees from $\hat{F}$, and with substantiated success. However, all of the existing methods are heuristics, leaving room for improvement. We identify one subproblem that, if solved very fast, allows us to do exact PPM-based tree inference for problems of relevant biological sizes. It is this subproblem, a projection problem in Eq. (3), that is our focus. We introduce the first non-iterative algorithm to solve this projection problem, and show that it is $74\times$ faster than different optimally-tuned iterative methods. We are also the first to show that a full-exact-enumeration approach to inferring $U$ and $M$ from $\hat{F}$ is possible, in our case, using a GPU and our algorithm to compute and compare the cost of all the possible trees that might explain the data $\hat{F}$. EXACT often outperforms the above tools, none of which does exact inference. Our paper is not about EXACT, whose development challenges and significance for biology go beyond solving our projection problem, and which is the focus of our future work.

Despite this difference in purpose, in this section we compare the performance of inferring trees from a full exact search over the space of all possible PPM models with the performance of a few existing algorithms. In Figure 5, we compare EXACT, PhyloWGS, CITUP and AncesTree on recovering the correct ancestry relations on biological datasets also used by [3]. A total of 30 different datasets [38], i.e., $\hat{F}$, were tested. We use the default parameters in all of the algorithms tested.

Figure 5: Comparison of different phylogenetic tree inference algorithms.

In each test, and for every pair of mutations $i$ and $j$, we use the tree output by each tool to determine if (a) $i$ is an ancestor of $j$ or if $j$ is an ancestor of $i$, (b) if $i$ and $j$ are in the same node, (c) if either $i$ or $j$ are missing in the tree, or, otherwise, (d) if $i$ and $j$ are incomparable. We give these four possible ancestral relations, the following names: *ancestral*, *clustered*, *missing*, and *incomparable*. A random guess correctly identifies 25% of the ancestral categories, on average. If the fraction of misidentified relations is 0, the output tree equals the ground-truth tree. All methods do better than random guesses.

For example, in Figure 6, according to EXACT, mutation 63, at the root, is an ancestor of mutation 57, at node 3. However, according to the ground truth, in Figure 6, they belong to the same node. So, as far as comparing 63 with 57 goes, EXACT makes a mistake. As another example, according tp EXACT, mutations 91 and 55 are incomparable, while according to the ground truth, 91 is a descendent of 55. Hence, as far as comparing 91 with 55 goes, EXACT makes another mistake. The fraction of errors, per ancestral relation error type, that each of these tools makes is: EXACT = $\{23\%, 10\%, 0\%, 13\%\}$; PhyloWGS = $\{3\%, 2\%, 0\%, 1\%\}$; AncesTree = $\{54\%, 16\%, 95\%, 25\%\}$; CITUP = $\{27\%, 13\%, 0\%, 21\%\}$.

In our experiments, EXACT performs, on average, better than the other three methods. PhyloWGS performs close to EXACT, however, it has a much longer run time. Although AncesTree does fairly well in terms of accuracy, we observe that it often returns trees with the same topology, a star-shaped tree. The other methods, produce trees whose topology seems to be more strongly linked to the input data. Finally, AncesTree's inferred tree does not cover all of the existing mutations. This behaviour is expected, as, by construction, AncesTree tries to find the largest tree that can be explained with the PPM. See Figure 6, and Figure 7, for an example of the output produced by different algorithms, and the corresponding ground truth.

Figure 6: Tree reconstructed by different algorithms for the first file in the folder [38]. AncesTree often outputs star-shaped trees. The small numbers listed next to each node represent mutations. Mutations indexed by the same number in different trees are the same real mutation. The root of each tree is circled in thick red. Nodes are labeled by numbers, and these labels are assigned automatically by each tool. Labels of different trees are incomparable.

We end this section by discussing a few extra properties that distinguished an approach like EXACT from the existing tools. Because our algorithm's speed allows a complete enumeration of all of the trees, EXACT has two unique properties. First, EXACT can exactly solve

$$\min_{U \in \mathcal{U}} \mathcal{J}(\mathcal{C}(U)) + \mathcal{Q}(U), \tag{72}$$

where $U$ encodes ancestral relations, $\mathcal{C}(U)$ is the fitness cost as defined in our paper, $\mathcal{J}$ is an arbitrary, fast-to-compute, 1D scaling function, and $\mathcal{Q}(U)$ is an arbitrary, fast-to-compute, tree-topology penalty function. No other tool has this flexibility. Second, EXACT can find the $k$ trees with the smallest objective value in (72). A few existing tools can output multiple trees, but only when these all have the same "heuristically-optimal" objective value. This feature is very important because, given that the input data is noisy, and the number of samples is often small, it allows, e.g., one to give a confidence score for the ancestry relations in the output tree. Furthermore, experiments show that the ground-truth tree can often be found among these $k$ best trees. Hence, using other biological principles, the ground-truth tree can often be identified from this set. Outputting just "heuristically-optimal" trees prevents this finding.

**Ground Truth**

Figure 7: Ground truth tree for the input file that generated Figure 6. The small numbers listed next to each node represent mutations. Mutations indexed by the same number in different trees are the same real mutation. The root of each tree is circled in thick red. Nodes are labeled by numbers, and these labels are assigned automatically by each tool. Labels of different trees are incomparable.