[Reviews · NeurIPS 2018]

Reviewer 1



The paper studies an algorithm to study efficient projection onto perfect phylogeny model. The major contribution of the paper is to give a Moreau's decomposition for proximal operator, and a tree reduction scheme to solve the projection problem. I find it hard to justify the novelty of the paper as the formulation is not too technical. The contribution of the paper should be from the application perspective. However, the paper did not provide sufficient numerical studies especially not using real data to provide real case studies. I thus do not suggest an acceptance. After reading the rebuttal, I still do not see enough new contribution from this paper. I think this should be a rejection,

Reviewer 2



This paper provides a very efficient algorithm to reconstruct phylogenetic models (trees). They key insights are efficient ways to search the entire subspaces around a proposed model, so that a full search can be made overall efficiently. The algorithm runs in O(q^2 p) steps for a genome of length q, and with p samples. Experiments show that the algorithm runs significantly faster than other search algorithms they implemented, a x74 speed-up! The paper is well-written, although fairly dense. Re REBUTAL: Thank you for the time you took to provide a detailed rebuttal. However, I wished you had ONLY addressed the main concerns from Reviewer #2. I felt lost in the point of most of the comments.

Reviewer 3



This paper attempts to solve an optimization subproblem which arises in a matrix factorization which arises in studying the phylogenetic evolutionary tree. In particular, the overall problem is given a matrix F, find the matrices U and M such that F=UM where U is a binary matrix and M is a positive matrix whose columns sum to 1. The subproblem is, given U, to find the least-squares solution F,M such that || \hat{F} - F || is minimized subject to the constraints that F = UM where \hat{F} is the measured matrix. The paper proposes an algorithm which can solve this subproblem directly in polynomial time (compared with iterative solvers) and the major contribution is the algorithm for computing this along with the theoretical proofs of its optimality. Beyond this analysis, practical techniques are proposed for evaluating the algorithm efficiently. The algorithm is then applied and compared against current state-of-the-art iterative solvers, specifically ADMM and PGD. Experimental results show that the proposed algorithm is dramatically faster in practice when compared against existing iterative solvers. Overall the paper seems well written and the results significant. I suspect the paper will be of interest beyond those studying phyolgenetic trees as the main contribution is related to the optimization sub problem. UPDATE: Having read the author feedback I remain largely positive on the paper. In addition, the authors included additional experimental comparison against other methods seems to be a very positive and compelling addition to the paper.

Reviewer 4



This is a nice mixed programming problem to a topical problem of inferring the most plausible perfect phylogeny in the setting of noisy data, that is the input matrix contains fractions of the population with mutations at the respected position. The authors propose a 2-stage procedure in which in the first stage, they find good candidate matrices U's from the space of valid trees, and in the second stage they find the exact goodness of that matrix/tree. The space of these matrices is exponential in the dimension of the matrix U (although these matrices are over 1/0 only) and therefore the emphasis is over this stage. The authors claim they achieve reasonable results for q=11 and they compared it to other implementations. I cannot see a real practical use for such magnitude, but I trust the authors they obtained good results.